# Efficient Computation of Deep Nonlinear ∞-Width Neural Networks That Learn Features

**Greg Yang** [*]
Microsoft

**Michael Santacroce**
Microsoft

**Edward J. Hu**
Microsoft

## Abstract

While a popular limit of infinite-width neural networks, the Neural Tangent Kernel (NTK) often exhibits performance gaps from finite-width neural networks on standard datasets, due to lack of feature learning. Although the feature learning *maximal update limit*, or $\mu$-*limit* (Yang and Hu, 2020) of wide networks has closed the gap for 1-hidden-layer linear models, no one has been able to demonstrate this for deep nonlinear multi-layer perceptrons (MLP) because of $\mu$-limit's computational difficulty in this setting. Here, we solve this problem by proposing a novel feature learning limit, the $\pi$-*limit*, that bypasses the computational issues. The $\pi$-limit, in short, is the limit of a form of *projected gradient descent*, and the $\pi$-limit of an MLP is roughly another MLP where gradients are *appended* to weights during training. We evaluate it on CIFAR10 and Omniglot against NTK as well as finite networks, finding the $\pi$-limit outperform finite-width models trained normally (without projection) in both settings, closing the performance gap between finite- and infinite-width neural networks previously left by NTK. Code for this work is available at `github.com/santacml/pilim`.

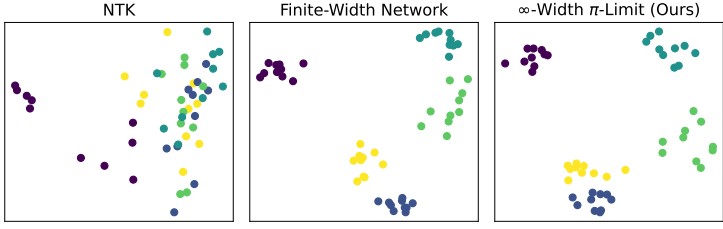

Figure 1: **PCA of representations of images from 5 classes (= 5 colors) in Omniglot test set.** We compare the representations from our best performing finite-width $\mu$-parametrized network, our proposed ∞-width $\pi$-limit, and NTK. The representations from former two form neat clusters according to their classes while those from the latter are mixed up together. See Appendix B.5.3.

## 1 Introduction

The theory of Neural Tangent Kernel (NTK) (Jacot et al., 2018) led to many theoretical discoveries of neural networks, but they ultimately apply only to a limited, unrealistic regime where the network does not learn features. Recently, Yang and Hu (2020) discovered the *maximal update, or $\mu$, parametrization* that induces a feature learning ∞-width limit, called the $\mu$-*limit* of deep neural networks, which generalizes the mean field limit of shallow networks (Sirignano and Spiliopoulos, 2018; Mei et al., 2018; Chizat and Bach, 2018; Rotskoff and Vanden-Eijnden, 2018). They explicitly trained such ∞-width limits on Word2Vec and MAML, tasks relying crucially on feature learning. These $\mu$-limits outperformed both NTK baselines and finite-width networks. Thus, the $\mu$-limit seems to capture key properties of neural networks in practice and is a promising direction to further our theoretical understanding of them.

However, a significant roadblock to this end is that the $\mu$-limit for general neural networks is both hard to compute empirically and hard to analyze theoretically. Indeed, Yang and Hu (2020) restricted main experiments to linear MLPs where exact calculation is fast as a special case. According to them, the main complications come from the interleaving of forward and backward propagations that leads to the need to evaluate Gaussian integrals involving nested nonlinearities. Likewise, these complicated integrals also contribute to the difficulty of theoretical analysis, especially in deep networks.

---

[*]Correspondence to {gregyang,Michael.Santacroce}@microsoft.com

**Contributions** In this work, we bypass these issues by considering a form of *projected gradient descent* instead. Roughly speaking, with $\mathcal{V}_\phi$ denoting the V-transform (aka dual) of $\phi$ (Definition 3.1).

$$\underset{\text{trained by \textit{projected} gradient \textit{accumulation}}}{\text{MLP with nonlinearity } \phi} \xrightarrow{width \to \infty} \underset{\text{trained by gradient \textit{concatenation}}}{\text{MLP with nonlinearity } \mathcal{V}_\phi} \qquad (\star)$$

Here *gradient accumulation* is just the normal training process where gradient is *added* to a parameter, whereas in *gradient concatenation*, gradient is *appended* to a parameter. The RHS retains the feature learning capability while making the computation and the mathematics much simpler compared to the $\mu$-limit. We explicitly evaluate our new limit, which we call the $\pi$-*limit*, on CIFAR10 and Omniglot, and compare against finite-width neural networks as well as NTK and NNGP. We find the $\pi$-limit always outperform wide standard-parametrized or $\mu$-parametrized networks, and they all significantly outperform the kernels. For example, on Omniglot, this is evidenced by Fig. 1. We also demonstrate transfer learning from ImageNet to CIFAR10. To our knowledge, this is the first time deep *nonlinear* feature learning $\infty$-width limits are evaluated on nontrivial natural datasets.

## 2 WHY IS THE $\mu$-LIMIT HARD TO COMPUTE? AN EXAMPLE

While the $\mu$-limit is the "optimal" feature learning limit of a neural network as described in Yang and Hu (2020), it is difficult to calculate analytically. Let's demonstrate this difficulty with a simple 1-hidden-layer $\mu$-parametrized network (abbreviated $\mu$-*net*, whose $\infty$-width limit is the $\mu$-*limit*) $f$ with 1-dimensional input $\xi \in \mathbb{R}$ and output $f(\xi) \in \mathbb{R}$:

$$f(\xi) = n^{-1/2} v^\top \phi(\sqrt{n} u \xi) \in \mathbb{R}, \qquad (1)$$

where $u, v \in \mathbb{R}^n$ are column vectors and $\phi : \mathbb{R} \to \mathbb{R}$ is the nonlinearity (e.g. relu). Note here $n$ is the width of $f$. According to $\mu$-parametrization, we initialize $u_\alpha, v_\alpha \sim \mathcal{N}(0, \frac{1}{n})$.

At initialization, one easily sees $\lim_{n \to \infty} f(\xi) = 0$ because

$$f(\xi) = \frac{1}{n} \sum_{\alpha=1}^n (\sqrt{n} v)_\alpha \phi(\xi \sqrt{n} u_\alpha)$$

is the average of a large number of iid random variables $(\sqrt{n} v)_\alpha \phi(\xi \sqrt{n} u_\alpha)$ with zero mean. In the language of Tensor Programs (Yang, 2019b;a; 2020a;b), we have an almost sure convergence

$$f(\xi) \to \mathbb{E}\, Z^{\sqrt{n} v} \phi(\xi Z^{\sqrt{n} u}) = 0, \quad \text{at initialization,}$$

where $Z^{\sqrt{n} v}$ and $Z^{\sqrt{n} u}$ represent the independent standard Gaussian random variables describing the coordinate distributions of $\sqrt{n} v$ and $\sqrt{n} u$.

However, one can see that $f(\xi)$ has a limit of the form

$$f(\xi) \to \mathbb{E}(Z^{\sqrt{n} v} + \cdots)\phi(\xi Z^{\sqrt{n} u} + (\cdots)\phi'(\cdots)), \quad \text{after 1 step of SGD,} \qquad (2)$$

where $u$ and $v$ still represent values at initialization, and $\cdots$ represent terms coming from the gradient update.[1] Even though the $\cdots$ appearing inside $\phi()$ turn out to be all Gaussians, **this expectation is hard to evaluate analytically due to the nesting of $\phi'$ inside $\phi$**, unless $\phi$ is polynomial. Furthermore, as more SGD steps are taken, this nesting quickly becomes more complicated, so even if $\phi$ is polynomial, the time needed for evaluation compounds exponentially.

## 3 SOLUTION: PROJECT THE GRADIENT

To alleviate this difficulty, we propose to study the limit of training a neural network under a projected form of gradient descent which we call $\pi$-**SGD**, i.e., we update $\theta \leftarrow \theta - \eta \Pi \nabla_\theta \mathcal{L}$ where $\Pi$ is a linear projection and $\theta$ is the vector of all parameters. We shall first describe this projection $\Pi$ for the simple motivating example above before doing so for the general case.

### 3.1 CONTINUING THE 1-HIDDEN-LAYER EXAMPLE

In the example of Eq. (1), $\Pi$ acts on the gradients as follows: 1) $\Pi$ leaves the output layer gradient $\nabla_v \mathcal{L}$ untouched, but 2) projects the input layer gradient[2] $\nabla_u \mathcal{L} = (\cdots)\phi'(\cdots)$ to the linear span of $u_0$, the initial value of input weights $u$:[3]

$$\Pi(\nabla_v \mathcal{L}, \nabla_u \mathcal{L}) = (\nabla_v \mathcal{L}, \Pi_{u_0} \nabla_u \mathcal{L}) = (\nabla_v \mathcal{L}, c u_0) \quad \text{for some } c \in \mathbb{R}.$$

---

[1] See Appendix C for a detailed calculation.

[2] The precise form is $\nabla_u \mathcal{L} = \mathcal{L}' v \phi'(\sqrt{n} u \xi_0)$.

[3] This projection may look excessively reductive, but this is just an artifact of the 1-dimensional input and the shallow depth. See Section 3.2 for the general case.

Here $\Pi_{u_0}$ denotes the orthogonal projection to the span of $u_0$. In particular, after projection, $\nabla_u \mathcal{L}$ now has roughly iid Gaussian coordinates (proportional to $u_0$).[4] Then unlike Eq. (2), we avoid the nesting of $\phi'$ inside $\phi$:

$$f(\xi) \to \mathbb{E}(Z^{\sqrt{n}v} + \cdots)\phi(\tilde{c}Z^{\sqrt{n}u}), \quad \text{after 1 step of } \pi\text{-SGD}, \tag{3}$$

for some deterministic scalar $\tilde{c}$. This expectation can then be evaluated routinely using the V-transform of $\phi$ (e.g. Cho and Saul (2009) for relu; see Definition 3.1 below), and likewise, so can the expectations involved in all later steps. After formalization inside a Tensor Program, this rigorously gives rise to the $\infty$-width limit of $f$ trained under $\pi$-SGD, which we call the $\pi$-**limit of** $f$. See Theorem 3.2.

**Definition 3.1.** Let $\mathcal{V}_\phi : \mathbb{R}^3 \to \mathbb{R}$ denote the *V-transform* of $\phi$, such that $\mathcal{V}_\phi(\mathbb{E}\,XY, \mathbb{E}\,X^2, \mathbb{E}\,Y^2) = \mathbb{E}\,\phi(X)\phi(Y)$ for any centered jointly Gaussian random variables $(X, Y) \in \mathbb{R}^2$. We will treat $\mathcal{V}_\phi$ like an activation function and automatically vectorize when $\mathcal{V}_\phi$ is applied to 3 vectors of the same length.

### 3.1.1 COMPUTING THE $\pi$-LIMIT

Here we sketch how to compute the $\pi$-limit of Eq. (1). Let $u_t, v_t, f_t$ denote corresponding objects at time $t$, with $t = 0$ denoting initialization.

**Memory and Time Requirements** What information do we need to store in order to calculate the $\pi$-limit? Because of the projection, $u_t = c_t u_0$ for some $c_t \in \mathbb{R}$. So to calculate the limit of $f_t$, we need to track $c_t$ (or rather its $\infty$-width limit). This memory requirement is $\Theta(1)$ in training time $t$. To see the form of $v_t$, it helps to simplify our setup a bit more by 1) initializing $v_0 = 0$ and by 2) assuming each $\pi$-SGD step involves a minibatch containing a sole input $\xi_t$. Correspondingly, because the gradient of $v_t$ is always proportional to $\phi(\xi_t\sqrt{n}u_t) = \phi(\xi_t c_t \sqrt{n}u_0)$, there exist coefficients $\{a_s \in \mathbb{R}\}_{s=0}^{t-1}, \{b_s \in \mathbb{R}\}_{s=0}^{t-1}$ (which are random, but deterministic conditioned on $u_0, v_0$) such that

$$\sqrt{n}v_t = \sum_{s=0}^{t-1} a_s \phi(b_s \sqrt{n}u_0). \tag{4}$$

Here $b_s = \xi_s c_s$ and $a_s$ is formed from the learning rate and loss derivative. So to calculate the limit of $f_t$, we need $\{a_s \in \mathbb{R}\}_{s=0}^{t-1}, \{b_s \in \mathbb{R}\}_{s=0}^{t-1}$ (or rather their $n \to \infty$ limits). Note that $c_t, a_s, b_s$ all have nontrivial fluctuations for finite $n$, but become deterministic as $n \to \infty$.

This implies a memory requirement of $\Theta(t)$, which is also the total requirement for computing the limit of $f_t$. As we will see, each forward and backward pass has runtime $\Theta(t)$ as well, so the total runtime for calculating $\lim_{n\to\infty} f_t$ is $\Theta(t^2)$,[5] compared to the exponential runtime of general $\mu$-limit.

**Forward Pass** Using Eq. (4), we can intuit (and formalize using Tensor Programs) that

$$f_t(\xi) = \frac{1}{n}\sum_{\alpha=1}^{n} \left(\sum_{s=0}^{t-1} a_s \phi(b_s\sqrt{n}u_{0\alpha})\right) \phi(\xi c_t \sqrt{n}u_{0\alpha}) \to \sum_{s=0}^{t-1} \mathring{a}_s \,\mathbb{E}\,\phi(\mathring{b}_s Z^{\sqrt{n}u_0})\phi(\xi\mathring{c}_t Z^{\sqrt{n}u_0}) \tag{5}$$

where $\mathring{a}_s, \mathring{b}_s, \mathring{c}_t$ denote the deterministic limits of the corresponding quantities. The expectation in the RHS can be evaluated using V-transforms. The $t$ terms in the summation implies a runtime of $\Theta(t)$.

**Backward Pass** The gradient update for the output weights $v_t$ is clearly represented by setting $(a_{t+1}, b_{t+1}) \leftarrow (-\eta\mathcal{L}', \xi c_t)$, where $\mathcal{L}'$ denotes the loss derivative $\partial\mathcal{L}(f_t(\xi), \text{label})/\partial f_t(\xi)$. However, a priori, it seems unclear how to calculate the limit of the *projected* gradient $\Pi_{u_0}\nabla_{u_t}\mathcal{L}$ of the input weights. Fortunately, because we can express the entire unrolled $\pi$-SGD training inside a Tensor Program, the Master Theorem (Yang, 2020b) can automatically give us the exact formulas for the gradient limit; see Appendix F.

But in fact, there is a more intuitive way of obtaining the backward pass limit by recognizing that a *projected gradient is just the maximal ascent direction in the projected space.*[6] In our case, $\Pi_{u_0}\nabla_{u_t}\mathcal{L}$

---

[4]This is because, while $c$ has fluctuations for finite $n$, it is roughly constant for large $n$.

[5]Assuming minibatch size is constant, and we train for a constant number of epochs, this translates to $\Theta(N^2)$ to train $\lim f_t$, where $N$ is dataset size, and $\Theta(N)$ to do inference on a single input. This compares favorably with Gaussian Processes, which requires $\Theta(N^3)$ to "train" (i.e. inverting the kernel), and $\Theta(N)$ for inference. However, in our experiments here, the constant in our $\Theta(N^2)$ in practice will make training $\lim f_t$ slower than the corresponding NTK or NNGP kernel regression.

[6]See Lemma F.1 for more details.

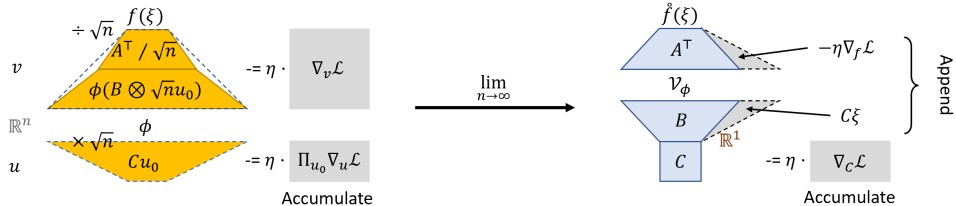

Figure 2: **Summary of the $\pi$-limit for the 1-hidden-layer network of Eq. (1).** *(Left)* Representation of width-$n$ $\mu$-net with nonlinearity $\phi$ and its $\pi$-SGD update. Here $v$ is depicted as $\frac{1}{\sqrt{n}}A^\top\phi(B\otimes \sqrt{n}u_0)$, which is equivalent to Eq. (4). In $\pi$-SGD, the first layer gradient is projected to the space spanned by $u_0$ before being accumulated to the weights. *(Right)* This network's $\infty$-width limit can be roughly thought of as another 1-hidden-layer neural network with nonlinearity $\mathcal{V}_\phi$ (Eq. (6)). But updates are *appended* to $A, B$ instead of added. See Theorem 3.2 and compare to Eq. ($\star$).

is just the maximal ascent direction of $\mathcal{L}$ in the span of $u_0$, i.e. $\Pi_{u_0}\nabla_{u_t}\mathcal{L} = (\nabla_{c_t}\mathcal{L})u_0$, where $\nabla_{c_t}\mathcal{L}$ is the derivative w.r.t. the coefficient $c_t$ of $u_t$ in terms of $u_0$. In the $\infty$-width limit, this derivative can be obtained by just auto-differentiating Eq. (5) when the V-transform has an explicit form, like for $\phi = relu$ (Cho and Saul, 2009). So a $\pi$-SGD step on $u_t$ is equivalent to taking $c_{t+1} \leftarrow c_t - \eta\nabla_{c_t}\mathcal{L}$.

**Summary** Below, for brevity, we say *training routine* to mean the package of learning rate $\eta$, training sequence of singleton minibatches $\{(\xi_t, y_t)\}_{t\geq 0}$ (where $\xi_t$ is the input and $y_t$ is the label),[7] and a loss function $\mathcal{L}(f(\xi), y)$ that is continuously differentiable in the prediction of the model $f(\xi)$.

**Theorem 3.2.** *Consider the simple motivating example of 1-hidden-layer network $f$ in Eq. (1) with 1-dimensional input and outputs, initialized by $u_\alpha \sim \mathcal{N}(0, 1/n)$ and $v_\alpha \leftarrow 0$. Suppose its nonlinearity $\phi$ has a polynomially bounded 2nd derivative. For any training routine, $f$ trained by $\pi$-SGD for $T$ steps has an $\infty$-width limit $\mathring{f}_T$, in the sense that,*

$$\text{as } n \to \infty, \quad f_T(\xi) \to \mathring{f}_T(\xi), \quad \text{for any } \xi \in \mathbb{R},$$

*where the limit is almost sure. $\mathring{f}_T$ is given as follows:*
*Initialize $A_0, B_0$ as empty column vectors, and initialize $C_0 \in \mathbb{R}$ as 1. Recall $\mathcal{V}_\phi$ is the V-transform of $\phi$. For each $t = 0, 1, \ldots$, define the function (where dot product of empty vectors is 0)*

$$\mathring{f}_t(\xi) \stackrel{def}{=} A_t^\top \mathcal{V}_\phi(C_t\xi B_t, B_t \circ B_t, C_t^2\xi^2\mathbf{1}), \quad \text{for any } \xi \in \mathbb{R}, \tag{6}$$

*where $\circ$ denotes coordinatewise product and $\mathbf{1}$ denotes the all-1s vector of appropriate shape, and $A_t, B_t, C_t$ are inductively given by $C_{t+1} \stackrel{def}{=} C_t - \eta\nabla_{C_t}\mathcal{L}(\mathring{f}_t(\xi_t), y_t)$ and*

$$A_{t+1} \stackrel{def}{=} append(A_t, -\eta\nabla_{\mathring{f}_t(\xi_t)}\mathcal{L}(\mathring{f}_t(\xi_t), y_t)), \qquad B_{t+1} \stackrel{def}{=} append(B_t, C_t\xi_t)$$

*where $append(v, p)$ appends element $p$ to the end of vector $v$, increasing the dimension of $v$ by 1.*

Here $A_t, B_t$ correspond to the column vectors formed from the $\infty$-width limits of $\{a_s \in \mathbb{R}\}_{s=0}^{t-1}, \{b_s \in \mathbb{R}\}_{s=0}^{t-1}$, and $C_t$ corresponds to the limit of $c_t$.[8] As discussed above, computing $\mathring{f}_T$ requires $\Theta(T)$ memory and $\Theta(T^2)$ time. Theorem 3.2 is summarized by Fig. 2.

## 3.2 $\pi$-PARAMETRIZATION FOR DEEP NETWORKS

We can straightforwardly generalize $\pi$-SGD and the $\pi$-limit theorem (Theorem 3.2) to deep MLPs. However, due to the $n \times n$ Gaussian matrix initialization in the middle of the network, the memory requirement will be $\Theta(T^2)$ and runtime will be $\Theta(T^3)$ for training $T$ steps, for the same reason as discussed in Yang and Hu (2020, Sec 8). This is not scalable to datasets like CIFAR10. Therefore, we propose a different initialization that brings down the memory requirement to $\Theta(T)$ and runtime to $\Theta(T^2)$, just like the 1-hidden-layer case. Consider an $L$-hidden-layer $\mu$-net $f : \mathbb{R}^d \to \mathbb{R}^{d_{\text{out}}}$: For

---

[7]For simplicity, we only consider batch size 1; it's straightforward to generalize to larger batch sizes.

[8]Technically, we should have written $\mathring{A}_t, \mathring{B}_t, \mathring{C}_t$ to maintain the convention that $\mathring{\square}$ denotes limit of $\square$, but for the sake of brevity, we drop this convention in Theorem 3.2.

weight matrices $w^1 \in \mathbb{R}^{n \times d}$ and $w^2, \ldots, w^L \in \mathbb{R}^{n \times n}$, and nonlinearity $\phi : \mathbb{R} \to \mathbb{R}$, such a neural network on input $\xi \in \mathbb{R}^d$ is given by $h^1(\xi) = \sqrt{n} w^1 \xi \in \mathbb{R}^n$, and

$$x^l(\xi) = \phi(h^l(\xi)) \in \mathbb{R}^n, \quad h^{l+1}(\xi) = w^{l+1} x^l(\xi) \in \mathbb{R}^n, \quad \text{for } l = 1, \ldots, L-1, \qquad (7)$$

and the network output is $f(\xi) = n^{-1/2} w^{L+1} x^L(\xi)$ for $w^{L+1} \in \mathbb{R}^{d_{\text{out}} \times n}$. In $\mu$-parametrization, we would initialize $w_{\alpha\beta}^l \sim \mathcal{N}(0, 1/n)$ for any $\alpha, \beta, l$. Next, we describe our alternative proposal.

$\pi$-**Initialization**  Choose integers $r, M$, to be explained shortly; these numbers should all be thought of as constant in width $n$. Suppose we are given a collection $\mathcal{P}$ of matrices

$$\mathcal{P} \overset{\text{def}}{=} \{A^l, B^l \in \mathbb{R}^{M \times r}\}_{l=2}^L \cup \{A^1 \in \mathbb{R}^{d \times r}\} \cup \{B^{L+1} \in \mathbb{R}^{M \times r}, A^{L+1} \in \mathbb{R}^{M \times d_{\text{out}}}\}. \qquad (8)$$

Then we can initialize weights $w^l$ by first sampling a standard random Gaussian matrix $\Omega \in \mathbb{R}^{n \times r}, \Omega_{\alpha i} \sim \mathcal{N}(0, 1)$, before setting

$$w^l \leftarrow \frac{1}{n} \Omega A^{l\top} \phi(B^l \Omega^\top) \in \mathbb{R}^{n \times n}, \text{for all hidden } l = 2, \ldots, L;$$

$$w^1 \leftarrow \frac{1}{\sqrt{n}} \Omega A^{1\top} \in \mathbb{R}^{n \times d}; \quad w^{L+1} \leftarrow \frac{1}{\sqrt{n}} A^{L+1\top} \phi(B^{L+1} \Omega^\top) \in \mathbb{R}^{d_{\text{out}} \times n} \qquad (9)$$

As an example, the initialization of the 1-hidden-layer case in Theorem 3.2 corresponds to $d = d_{\text{out}} = r = 1, M = 0$ and $A^1 = 1$. See Fig. 3(Left) for an illustration.

Let's digest this initialization scheme: 1) Of course, $A^l, B^l$ need to be initialized themselves, and in this paper we will just do so with Gaussian random initialization; see Appendix B.1 for the pseudo-algorithm. 2) $A^l, B^l, l \in [2, L]$, will play the same roles as $A, B$ in the 1-hidden-layer example above, whereas $A^1$ plays the same role as $C$ there. 3) $r$ is a measure of dimension of the the projection. More precisely, we will project hidden weight gradients from $\mathbb{R}^{n \times n}$ to $\mathbb{R}^{n \times r}$; see below. 4) $A^l, B^l$ will grow in the $M$ dimension with training time, just like how the sizes of $A, B$ grow in the 1-hidden-layer example. 5) Eq. (9) can be interpreted as saying $w^l$ are generated by 1-hidden-layer MLPs $F_{A^l, B^l} : \mathbb{R}^r \to (\mathbb{R}^r \text{ or } \mathbb{R}^{d_{\text{out}}})$ with weights $A^l, B^l$, like so:

$$w_{\alpha\beta}^l \leftarrow 1/n \langle \Omega_\alpha, F_{A^l, B^l}(\Omega_\beta) \rangle, \text{for all hidden } l = 2, \ldots, L; \quad w_{:\beta}^{L+1} \leftarrow 1/n F_{A^{L+1}, B^{L+1}}(\Omega_\beta). \qquad (10)$$

This is reminiscent of hypernetworks (Ha et al., 2016), but the crucial difference is that the "batch dimension" (indexed by $\alpha, \beta$) of the generator $F_{A^l, B^l}$ becomes the width dimension of the generated network $f$, so that the same generators can generate networks of arbitrary width.

$\pi$-**Projection**  We leave the output weights $w^{L+1}$ alone, but for any $w \in \{w^1, w^2, \ldots, w^L\}$, where $w$ has shape $n \times n$ or $n \times d$, we project the gradient $\nabla_w \mathcal{L}$ of the same shape by left multiplying by $\Pi_\Omega$, the projection matrix to the $r$-dimensional space spanned by the columns of $\Omega$. This means that preactivations $h^l$ are always in this space. However, the input side of $w$ is not projected, so we are optimizing the hidden weights in a $(r \times n)$-dimensional space, which still becomes infinite-dimensional as $n \to \infty$. We refer to SGD with $\pi$-projected gradients as $\pi$-**SGD**.

This brings us to the following

**Definition 3.3** ($\pi$-Parametrization and $\pi$-Limit). We define $\pi$-*parametrization* as the package of $\pi$-initialization and $\pi$-SGD,[9] where $r$ and the initial $M$ are clear from context. A network in $\pi$-parametrization is abbreviated $\pi$-*network or $\pi$-net*, and we define $\pi$-*limit* as its $\infty$-width limit.

### 3.2.1  COMPUTING THE $\pi$-LIMIT OF DEEP MLP

Here we show the $\pi$-limit forward and backward passes can be efficiently computed.

$\pi$-**Limit Forward Pass**  Let's write $f^{\mathcal{P}}$ for the (random) neural network $\pi$-initialized by $\mathcal{P}$ as in Eqs. (8) and (9). Then it's straightforward to see $f^{\mathcal{P}}$ has a deterministic limit $\mathring{f}^{\mathcal{P}}$ expressible as a composition of matrix multiplications and V-transforms:

**Theorem 3.4** ($\pi$-Limit Forward Pass). *Suppose $\phi$ has a polynomially bounded 2nd derivative. Then*

$$\text{as } n \to \infty, \quad f^{\mathcal{P}}(\xi) \to \mathring{f}^{\mathcal{P}}(\xi), \quad \text{for any } \xi \in \mathbb{R}^d,$$

*where convergence is almost sure, and $\mathring{f}^{\mathcal{P}}(\xi)$ is defined as follows. Write $g^1 \overset{\text{def}}{=} A^{1\top} \xi \in \mathbb{R}^r$,*

$$g^l \overset{\text{def}}{=} A^{l\top} \mathcal{V}_\phi(B^l g^{l-1}, B^l \circ B^l, \|g^{l-1}\|^2 \mathbf{1}) \in \begin{cases} \mathbb{R}^r & \text{for } l = 2, \ldots, L \\ \mathbb{R}^{d_{\text{out}}} & \text{for } l = L+1, \end{cases}$$

*and $\mathring{f}^{\mathcal{P}}(\xi) \overset{\text{def}}{=} g^{L+1}$, where $B \circ B$ yields a size-$M$ column vector of squared norms of $B$'s rows.*

---

[9] with learning rate independent of width (but may vary with training time)

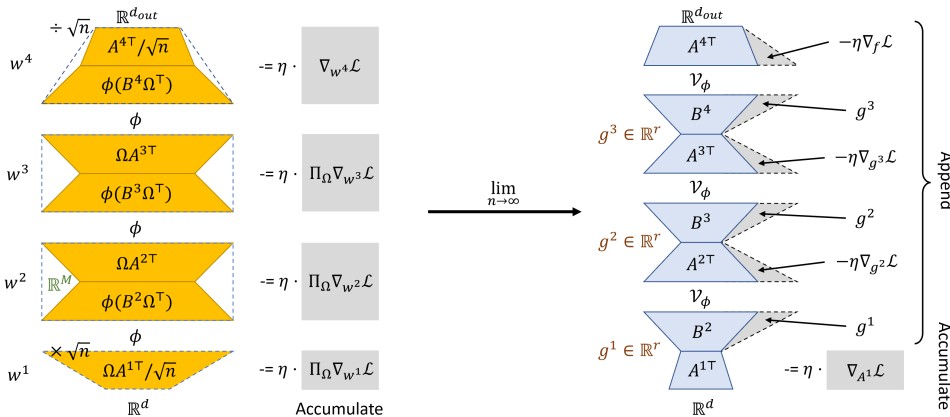

Figure 3: **Summary of the $\pi$-limit for deep MLPs Eq. (7),** in the same style as Fig. 2. Here we take the example of 3-hidden-layer MLPs. See Theorems 3.4 and 3.5 and compare to Eq. ($\star$).

See Fig. 3(Right) for an illustration. Here, $g^l$ represents the coefficients of preactivation $h^l$ (c.f. Eq. (7)) in the columns of $\Omega$. As we will see next, $\pi$-projection ensures that at any point during training, $f_t$ has an $\infty$-width limit of the form $\mathring{f}^{\mathcal{P}}$ for some $\mathcal{P}$.

**$\pi$-Limit Backward Pass** Just like the 1-hidden-layer case, we can interpret the projected gradient as the maximal ascent direction in the column space of $\Omega$. Formalizing the intuition using Tensor Programs yields the following theorem that we also empirically verify in Appendix E.

**Theorem 3.5** ($\pi$-Limit Backward Pass). *Let $\mathcal{P}_0$ denote the matrices (c.f. Eq. (8)) used in the $\pi$-initialization of $f$ as in Eq. (9). Suppose its nonlinearity $\phi$ has a polynomially bounded 2nd derivative. Then for any training routine, $f$ trained by $\pi$-SGD for $T$ steps has an $\infty$-width limit which is equal to $\mathring{f}^{\mathcal{P}_T}$ for some $\mathcal{P}_T$ given below, i.e.*

$$as\ n \to \infty, \quad f_T(\xi) \to \mathring{f}^{\mathcal{P}_T}(\xi), \quad for\ any\ \xi \in \mathbb{R}^d,$$

*where the limit is almost sure. $\mathcal{P}_T$ is given inductively through its elements $A_T^l, B_T^l$ as follows: $A_{t+1}^1 \stackrel{def}{=} A_t^1 - \eta \nabla_{A^1} \mathcal{L}(\mathring{f}^{\mathcal{P}_t}(\xi_t), y_t)$, and, for $l = 2, \ldots, L+1$,*

$$A_{t+1}^l \stackrel{def}{=} append(A_t^l, -\eta \nabla_{g_t^l} \mathcal{L}(\mathring{f}^{\mathcal{P}_t}(\xi_t), y_t)), \qquad B_{t+1}^l \stackrel{def}{=} append(B_t^l, g_t^{l-1}).$$

*Here $g_t^l$ corresponds to $g^l$ in Theorem 3.4 evaluated for input $\xi_t$ and function $\mathring{f}^{\mathcal{P}_t}$, and $append(B, g)$ means appending $g$ as a new row vector of $B$, increasing by 1 the dimension represented by $M$.*

Theorems 3.4 and 3.5 are summarized by Fig. 3.[10] As conveyed by Eq. ($\star$), the $\pi$-limit can be thought of as another MLP with activation $\mathcal{V}_\phi$ and trained by gradient concatenation.[11] Because $\pi$-parametrization has similar scaling with width as $\mu$-parametrization, it is easy to show that the former admits feature learning in the same way the latter does, i.e. the feature kernel of every layer evolves nontrivially during training. Using Tensor Programs, it is straightforward to extend the above theorems to more general settings such as biases, large batch size, per layer learning rate, or metalearning (Appendix A). Appendix D also makes several observations on the $\pi$-limit theorems.

**$\pi$-Limit vs $\mu$-Limit** While the projection means optimization is slower in the $\pi$-limit, we believe with sufficiently large $r$, the $\pi$-limit should perform similarly to the $\mu$-limit. Indeed, prior works such as Li et al. (2018) has shown that optimizing a neural network in a much smaller, *random* parameter subspace can recover most of the performance of training all parameters. This is also evidenced by our experimental results below, where the $\pi$-limit generally only slightly outperform wide $\mu$-networks.

---

[10]One may wonder how would Theorem 3.5 compare with just directly accumulating gradients on $\mathcal{P}$. In fact, this would train very badly because $\mathcal{V}_\phi$ is very smooth around 0, so $\mathring{f}^{\mathcal{P}}$ would look very linear for a long time. Empirically, direct SGD on $\mathcal{P}$ would yield $\leq 53\%$ test and $\leq 80\%$ training accuracy on CIFAR10.

[11]Note that once a vector is appended to $A^l$ or $B^l$, it is not touched again. Therefore, despite the similarity between the $\pi$-limit and an MLP, $A^l$ and $B^l$ should *not* be thought of "parameters" in the usual sense.

Table 1: **Best Test Accuracies on CIFAR10 and Omniglot**, best of MLPs up to 4 hidden layers, width 2048, $r$ 400, as well as random search over a host of hyperparameters; see Appendix B. Note the $\mu$-*Net* numbers are also the optimal numbers for *standard parametrization* finite networks, as discussed in Footnote 12. $\pi$-*Limit ImageNet Transfer* means pretraining a $\pi$-limit with $r = 200$ on ImageNet32 and perform kernel regression with its feature kernel (i.e. kernel of last layer activations) on CIFAR10; see Section 4.2 and Appendix B.4 for details and finite network results. Also compare with feature kernel regression without pretraining (Table 8).

|  | NNGP | NTK | $\substack{NTK \\ perf\ gap}$ | $\mu$-Net | $\pi$-Net | $\pi$-Limit | $\substack{\pi\text{-Limit} \\ \text{ImageNet Transfer}}$ |
|---|---|---|---|---|---|---|---|
| CIFAR10 | 58.92 | 59.63 | $\longleftrightarrow$ | 61.31 | 60.64 | **61.50** | **64.39** |
| Omniglot | 43.80 | 51.72 | $\longleftrightarrow$ | 91.22 | **92.21** | **91.46** | - |

## 4 EXPERIMENTS

Here we compare the performance of the relu $\pi$-limit on CIFAR10 (Krizhevsky, 2009) and Omniglot (Lake et al., 2015) against that of NTK, NNGP, and finite-width $\pi$- and $\mu$-nets.[12] As we will see, 1) $\pi$-limits of large enough $r$ beat finite $\mu$-nets; 2) finite $\pi$-nets underperform the above on CIFAR10 but, interestingly, *outperform* them on Omniglot; 3) all of the above beat NTK and NNGP.

### 4.1 CLASSIFICATION ON CIFAR10

**Experimental Setup** For $\pi$- and $\mu$-networks, infinite or finite, we train for 50 epochs. We adopt a step learning rate schedule, with a learning rate drop of 0.15 at a certain milestone, which is a hyperparameter. We sweep over a variety of hyperparameters such as the learning rate, gradient clipping, weight decay, the LR drop milestone, etc, as well as width, $r$, and depth. For NTK and NNGP, we perform kernel regression following Lee et al. (2018) using centered labels. For them, we sweep over the initialization variances and learning rate multipliers, along with ridge coefficient. See Appendix B for more details.

**Results** In the literature, the performance gap between CNN and its NTK (Arora et al., 2019) is often cited for the deficiency of the NTK theory for explaining neural networks in practice. However, in fact, on MLPs we already see a nontrivial gap, as seen in Table 1 (compare $\mu$-*Net* with *NTK*, *NNGP*).[13] The $\pi$-limit closes this gap, having almost a 2-point advantage over NTK. The finite-width $\pi$-net outperforms NTK and underperforms $\pi$-limit both by about 1 point.

**Feature Learning in $\pi$-Limit and Finite Networks** We show the advantage of $\pi$-limit over NTK and NNGP is due to feature learning. To do so, we track the kernel regression performance of its (last-layer) feature kernel over the course of training. As seen in Fig. 4(Left), while the feature kernel at initialization underperforms both NNGP and NTK, it improves consistently over time to eventually exceed them.[14] Similarly, the kernel performance of the best $\pi$- and $\mu$-nets improves over time as well, though the accuracy of the feature kernel regression is slightly less than the network itself, likely due to the low rank property of the feature kernel. On the contrary, the $\pi$-limit benefits from feature kernel regression, improving the test accuracy from 61.5% to 61.85%; see Table 8.

**Effects of Width, $r$, and Depth** As shown in Fig. 4(Middle), accuracy of $\pi$-net increases monotonically with width and with $r$, approaching the accuracy of $\pi$-limit and $\mu$-net, respectively, from below. This can also be seen in feature kernel regression across training time, Fig. 6. In contrast, performance is not monotonic in depth for any of $\mu$-net, $\pi$-net, or $\pi$-limit, beyond a consistent improvement from 1 to 2 hidden layers; see Fig. 7.

---

[12]Because 1) standard parametrization differs from $\mu$-parametrization only in factors depending on width, 2) we sweep such factors in hyperparameter optimization, and 3) our finite networks have width at most 2048 (so these factors are in practice constants), our best accuracies for $\mu$-nets will also be best accuracies for standard-parametrized MLPs. We also train $\pi$-nets with *untied* $\Omega$s; see Appendix A.7.

[13]Note that, contrary to Lee et al. (2018; 2020), which claimed that finite-width neural networks underperform the kernels, here we find the former *outperform* the latter. This is primarily due to the single learning rate drop we adopted, while Lee et al. (2018; 2020) used a constant learning rate. We believe this provides a fairer comparison to the kernels since 1) the step LR schedule is more common than constant learning rate in practice, and 2) the kernels would obtain the same performance if we did kernel gradient descent for infinite-time (which is equivalent to kernel regression) with learning rate drop, since this optimization is convex.

[14]In both NNGP and NTK limits, the feature kernel stays fixed throughout training (Yang and Hu, 2020), so the dotted line in Fig. 4(Left) shows the result if we do the same experiments for NNGP and NTK.

Figure 4: **(Left)** Feature kernels of $\pi$-limit, $\pi$-net, and $\mu$-net all improve in quality with training, as measured by kernel regression on CIFAR10. All models are the best from our hyperparameter sweeps, but note that feature kernel regression causes accuracy decrease in finite models. **(Middle)** CIFAR10 validation accuracy is monotonic across width and $r$ for $\pi$-net, $\pi$-limit, and $\mu$-net. **(Right)** Omniglot validation accuracy, in contrast, is not monotonic in width, and $\mu$-net can underperform $\pi$-net of large $r$. (Note we did not run $\pi$-limit with $r \geq 800$ in consideration of computational costs). All numbers in the heatmaps are the best from our random hyperparameter searches for 2-hidden-layer networks.

## 4.2 Transfer Learning from ImageNet to CIFAR10

Pretraining-and-transfer-learning is an important setting where feature learning is crucial. As pointed out in Yang and Hu (2020), the NTK limit trivializes pretraining, while the $\mu$-limit both theoretical and empirically benefit from it. Here we investigate pretraining for the $\pi$-limit in the image domain by pretraining on ImageNet32 (Chrabaszcz et al., 2017)[15] and transferring to CIFAR10.

**Experimental Setup** We pretrain the $\pi$-limit with $r = 200$ (as well as $\mu$-Net, $\pi$-Net with $r = 200$, and $\pi$-Net with $r = 400$) for 30 epochs on a fixed subset of ImageNet32 with 250 (out of 1000) randomly subsampled classes. To evaluate on CIFAR10, we compute the kernel induced by the pretrained final-layer-features and perform kernel regression.

**Results** As shown in Table 1, ImageNet32 pretraining nontrivially raises the downstream CIFAR10 performance over without pretraining, altogether creating a $5\%$ gap compared to NTK (which has the same test accuracy whether or not it is pretrained on ImageNet32 because of the disparate classes, as shown in Yang and Hu (2020)). This again demonstrates the feature learning capability of the $\pi$-limit. Table 7 shows the benefit of pretraining seems to be directly related to the capacity of the model.

## 4.3 Few-Shot Learning via Metalearning on Omniglot

Following Yang and Hu (2020), we also evaluate few-shot learning on Omniglot. Compared to traditional classification settings like CIFAR10, doing well on Omniglot requires learning features that can rapidly be adapted to new unseen data (Raghu et al., 2019). We will adopt a metalearning approach to this, following Finn et al. (2017).

Unlike Yang and Hu (2020), we will train our models using ANIL (*Almost No Inner Loop*) (Raghu et al., 2019) rather than first-order MAML (*Model Agnostic Meta-Learning*). Briefly, ANIL is a variant of second-order MAML, where in the inner loop, we only adapt the output layer, while in the outer loop, we only train the network body. We adopt ANIL because: 1) We would like to train deep MLPs with SGD and without Adam or batchnorm, but empirically this makes optimization of first-order MAML difficult for both finite and $\infty$-width networks compared to second-order MAML. 2) While a $\pi$-parametrized network trained by second-order MAML has an $\infty$-width limit calculable using Tensor Programs, it does not stay in the $f^{\mathcal{P}}$ form of Theorem 3.4 because of the second-order gradient, rendering the limit computation inefficient. 3) Fortunately, $\pi$-networks trained by ANIL do not have this issue because the inner and outer loop gradients are on different parameters. In addition, ANIL performs on par with MAML on standard finite-width networks (Raghu et al., 2019). 4) Furthermore, ANIL training more clearly delineates the role of the network body for learning reusable features and the role of the head for adapting to new data, which is more fitting for our goals in this paper.

**Experimental Setup** We focus on the 5-way, 1-shot task[16], with only 1 step of ANIL adaption. For $\pi$- and $\mu$-networks, infinite or finite, we train for 50 epochs, 1000 batches per epoch, and 8 tasks per batch. Following Antoniou et al. (2019), we use cosine annealing schedule on the meta-learning rate.

---

[15]i.e. ImageNet (Deng et al., 2009; Russakovsky et al., 2015) downsampled to $32 \times 32$ resolution
[16]i.e. each task consists of 5 different classes, with 1 example provided for each class in training.

For NTK and NNGP, ANIL meta-training has no effect, and meta-testing amounts to just taking 1 kernel gradient descent step on each task.[17] We sweep over a variety of hyperparameters such as the outer and inner learning rates, gradient clipping, etc, as well as width, $r$, and depth. See Appendix B for more details.

**Results**    As seen in Table 1, $\mu$-net, $\pi$-net, and $\pi$-limit all outperform NNGP and NTK by about 40 points. Interestingly, while $\pi$-limit is slightly better than $\mu$-net, they are both outperformed by $\pi$-net. This is related to the width nonmonotonicity seen on Omniglot, as we describe next.

**Effect of Width, $r$, and Depth**    While $\pi$-net performance with $r$ is still roughly monotonic, in contrast to in CIFAR10, here it can decrease with width, as seen in Fig. 4(Right). In addition, $\mu$-net seems to underperform $\pi$-net of the same width for sufficiently large $r$. We find this is primarily due to optimization and not generalization because Omniglot is hard to overfit; see Fig. 8. Counterintuitively, $\pi$-net of large $r$ can optimize faster than $\mu$-net despite the projection. This is likely a side effect of $\pi$-initialization, as we see this advantage persist when $r > width$, where projection is a no-op. We do not see width nonmonotonicity on CIFAR10 because overfitting is much easier there, so the results depend much more on the implicit bias of each model.

On the other hand, performance seems to monotonically increase along diagonals of fixed $r/width$ ratio, peaking around $r/width \approx 1/2$. This suggests a different limit of $r \propto width \to \infty$, which warrants investigation in future works.

Unlike CIFAR10, test accuracy is more monotonic in depth here (Fig. 7). Again, this is probably because the extra weights help with optimization, which is the bottleneck on Omniglot.

## 5    RELATED WORKS

The $\pi$-limit bears some superficial similarities to hierarchical kernel processes such as deep GPs (Damianou and Lawrence, 2013; Salimbeni and Deisenroth, 2017) or deep kernel processes (Aitchison et al., 2021). The crucial difference here is that 1) during inference, the $\pi$-limit is deterministic and not a hierarchically sampled process, and 2) the $\pi$-limit is not trained variationally.

Relatedly, deep GPs or kernel processes can be construed as a kind of randomly initialized infinite-width neural network with finite bottleneck layers (Agrawal et al., 2020; Aitchison, 2020). However, their variational inference procedures do not correspond to the limit of SGD. In contrast, Littwin et al. (2021) derived such an SGD limit where the network between consecutive bottlenecks is in NTK parametrization, and the limit takes the form of a kind of intertwined system of kernel gradient descent. However, the computation of this limit is not scalable to datasets like CIFAR10.

During training, the $\pi$-limit's $A^l$ and $B^l$ grow in size, reminiscent of works on growing neural networks (Liu et al., 2019; Wu et al., 2021; Gu et al., 2021; Gong et al., 2019). Of course, these works focus on improving training efficiency of practical models, which is not our goal here. Nevertheless, it will be interesting to explore whether insights from $\pi$-limit can contribute back to this literature.

Our $\pi$-initialization (given fixed $\mathcal{P}$) can be construed as a special form of the initialization in deep mean field limits of MLPs (Araújo et al., 2019; Sirignano and Spiliopoulos, 2020; Fang et al., 2020; Nguyen, 2019; Nguyen and Pham, 2020). However, without $\pi$-projection, these limits' analytical computation suffers exponential runtime blowup like the $\mu$-limit.

## 6    CONCLUSION

A good model for studying wide neural networks should 1) capture the desired behavior of neural networks in practice, including feature learning and performance on real datasets, and 2) come with a set of theoretical tools for working researchers. But an *attractive* model must also balance these properties with 3) computational simplicity for empirical investigations, and 4) mathematical simplicity for theoretical investigations.

Previously, NTK and NNGP satisfy 2), 3), 4) but not 1), as shown in this and prior works. The $\mu$-limit satisfies 1) and 2) but, arguably, not 3) and 4). In this work, we presented the $\pi$-limit which, we believe, satisfies all 4 properties and that can prove fruitful for future researches.

While our work closed the $\infty$-width performance gap, it also opens many new questions, e.g., Can we derive the $\pi$-limit for a fixed $r/width$ ratio? When precisely does the width nonmonotonicity of $\pi$-net occur? What about modern architectures, beyond MLP? We leave these questions to future work.

---

[17]NTK and NNGP can be trained under second-order MAML, but we found their performance strictly decreases with metatraining, as the randomization of labels across tasks confuse the readout layer.

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

# A   EXTENSIONS OF $\pi$-LIMIT

## A.1   BIASES

Biases are straightforward to add.

Consider an $L$-hidden-layer $\mu$-net $f : \mathbb{R}^d \to \mathbb{R}^{d_{\text{out}}}$ with biases: such a neural network on input $\xi \in \mathbb{R}^d$ is given by $h^1(\xi) = \sqrt{n}w^1\xi + \sqrt{n}b^1 \in \mathbb{R}^n$, and

$$x^l(\xi) = \phi(h^l(\xi)) \in \mathbb{R}^n, \quad h^{l+1}(\xi) = w^{l+1}x^l(\xi) + \sqrt{n}b^l \in \mathbb{R}^n, \quad \text{for } l = 1, \ldots, L-1, \quad (11)$$

and the network output is $f(\xi) = n^{-1/2}w^{L+1}x^L(\xi) + b^{L+1}$ for $w^{L+1} \in \mathbb{R}^{d_{\text{out}} \times n}$. In $\mu$-parametrization, the biases are initialized as $b_\alpha^l \sim \mathcal{N}(0, \sigma_b^2/n)$ for $l = 1, \ldots, L$, and $b_i^{L+1} \sim \mathcal{N}(0, \sigma_b^2)$, for a hyperparameter $\sigma_b$. However, in $\pi$-parametrization, we will do this a bit differently.

$\pi$-**Initialization**   Suppose we are given a collection $\mathcal{P}$ of matrices and vectors

$$\mathcal{P} \overset{\text{def}}{=} (\text{RHS of Eq. (8)}) \cup \{\beta^l \in \mathbb{R}^r\}_{l=1}^L \cup \{\beta^{L+1} \in \mathbb{R}^{d_{\text{out}}}\}. \quad (12)$$

Then we can initialize weights and biases by first sampling a standard random Gaussian matrix $\Omega \in \mathbb{R}^{n \times r}, \Omega_{\alpha i} \sim \mathcal{N}(0,1)$, and 1) set $w^l$ as in Eq. (9), and 2) set biases as

$$b^{L+1} \leftarrow \beta^{L+1} \quad \text{and} \quad b^l \leftarrow \frac{1}{\sqrt{n}}\Omega\beta^l \quad \text{for all } l = 1, \ldots, L. \quad (13)$$

This constitutes the $\pi$-initialization of biases.

$\pi$-**Projection**   For $\pi$-projection, we also project the bias gradients by $\Pi_\Omega$.

$\pi$-**Limit Calculation**

**Theorem A.1** ($\pi$-Limit Forward Pass with Bias). *Let $\mathcal{P}$ be some collection as in Eq. (12). As $n \to \infty$,*

$$f^{\mathcal{P}}(\xi) \to \mathring{f}^{\mathcal{P}}(\xi), \quad \text{for any } \xi \in \mathbb{R}^d,$$

*where convergence is almost sure, and $\mathring{f}^{\mathcal{P}}(\xi)$ is defined as follows. Write $g^1 \overset{\text{def}}{=} A^{1\top}\xi + \beta^1 \in \mathbb{R}^r$,*

$$g^l \overset{\text{def}}{=} A^{l\top}\mathcal{V}_\phi(B^l g^{l-1}, B^l \circ B^l, \|g^{l-1}\|^2 \mathbf{1}) + \beta^l \in \begin{cases} \mathbb{R}^r & \text{for } l = 2, \ldots, L \\ \mathbb{R}^{d_{\text{out}}} & \text{for } l = L+1, \end{cases}$$

*and $\mathring{f}^{\mathcal{P}}(\xi) \overset{\text{def}}{=} g^{L+1}$, where $B \circ B$ yields a size-$M$ column vector of squared norms of $B$'s rows.*

**Theorem A.2** ($\pi$-Limit Backward Pass). *For the same setting as in Theorem 3.5 but with $\mathcal{P}$ as in Eq. (12), $A^l, B^l$ are updated exactly as in Theorem 3.5, and $\beta^l$ is updated by gradient accumulation like $A^1$:*

$$\beta_{t+1}^l \overset{\text{def}}{=} \beta_t^l - \eta \nabla_{\beta^l} \mathcal{L}(\mathring{f}^{\mathcal{P}_t}(\xi_t), y_t), \quad \text{for all } l = 1, \ldots, L+1$$

## A.2   PARAMETER MULTIPLIERS

We can insert to Eq. (11) constant parameter multipliers $\alpha_w$ for each parameter $w$ like so

$$h^{l+1}(\xi) = \alpha_{w^{l+1}}w^{l+1}x^l(\xi) + \sqrt{n}\alpha_{b^l}b^l \in \mathbb{R}^n, \quad \text{for } l = 1, \ldots, L-1, \quad (14)$$

These multipliers are tuneable hyperparameters. They affect both the forward and backward passes of the network.

In the $\pi$-limit, the forward pass is the same as in Theorem A.1 except we replace $A^l$ with $\alpha_{w^l}A^l$ and $\beta^l$ with $\alpha_{b^l}\beta^l$. The backward pass is the same as in Theorem A.2, but we just have to make sure that we backprop through the multipliers (in contrast, if we instead have absorbed the multipliers into the initialization, then we would not backprop through the multilpliers).

In our experiments, we only consider the input weight multiplier, output weight multiplier, and a single multiplier for all biases.

### A.3 LEARNING RATE MULTIPLIERS

We may have custom learning rates for specific weights or biases. In our experiments, we implement this with learning rate multipliers relative to the global learning rate $\eta$, e.g., the learning rate of a parameter $w$ becomes $\gamma_w \eta$ if the multiplier is denoted $\gamma_w$. Then in the limit, we just need to replace $\eta$ in Theorem 3.5 or Theorem A.2 with $\gamma_{w^l} \eta$, i.e.

$$A_{t+1}^1 \stackrel{\text{def}}{=} A_t^1 - \gamma_{w^1} \eta \nabla_{A^1} \mathcal{L}(\mathring{f}^{\mathcal{P}_t}(\xi_t), y_t)$$

$$A_{t+1}^l \stackrel{\text{def}}{=} append(A_t^l, -\gamma_{w^l} \eta \nabla_{g_t^l} \mathcal{L}(\mathring{f}^{\mathcal{P}_t}(\xi_t), y_t)), \qquad \text{for all } l = 2, \ldots, L+1$$

$$\beta_{t+1}^l \stackrel{\text{def}}{=} \beta_t^l - \gamma_{b^l} \eta \nabla_{\beta^l} \mathcal{L}(\mathring{f}^{\mathcal{P}_t}(\xi_t), y_t), \qquad \text{for all } l = 1, \ldots, L+1.$$

But note we do not modify the $B^l$ update. In our experiments, we will only sometimes use a LR multiplier on the input layer ("Input Layer LR Mult"), one on the output layer ("Output Layer LR Mult"), and/or a single multiplier for all biases ("Bias LR Mult").

### A.4 LARGE BATCH SIZE

For batch size $S > 1$, we still accumulate gradients into $A^1$ and $\beta^l$ as if they are regular parameters, and for the hidden weights, we just append the $S$ gradient vectors as if they are from $S$ unit-sized batches.

### A.5 GRADIENT CLIPPING

The Frobenius norms of the projected gradients of the input weight $w^1$ and all biases $b^l$ converge to exactly the Frobenius norms of the gradients of $A^1$ and $\beta^l$, i.e.

$$\|\Pi_\Omega \nabla_{w^1} \mathcal{L}\|_F \to \|\nabla_{A^1} \mathcal{L}\|_F$$

$$\|\Pi_\Omega \nabla_{b^l} \mathcal{L}\| \to \|\nabla_{\beta^l} \mathcal{L}\|$$

where the convergence is almost sure. For any hidden weights $w^l$, suppose its gradient over an $S$-sized batch in the $\pi$-limit is given by

$$A_{t+1}^l \stackrel{\text{def}}{=} append(A_t^l, -\eta \tilde{A}), \qquad\qquad B_{t+1}^l \stackrel{\text{def}}{=} append(B_t^l, \tilde{B}).$$

where $\tilde{A}, \tilde{B} \in \mathbb{R}^{S \times r}$, and $append(B, \tilde{B})$ means appending all $S$ rows of $\tilde{B}$ into $B$, increasing the latter's column length by $S$. Then

$$\|\Pi_\Omega \nabla_{w^l} \mathcal{L}\|_F \to \sqrt{\text{tr}\left(\tilde{A}^\top \tilde{\phi}(\tilde{B}\tilde{B}^\top)\tilde{A}\right)}$$

where the convergence is almost sure. Then clipping the gradient in the $\pi$-limit just means clipping the gradients of $A^1$ and $\beta^l$, and, for hidden weights, rescale $\tilde{A}$ such that $\sqrt{\text{tr}\left(\tilde{A}^\top \tilde{\phi}(\tilde{B}\tilde{B}^\top)\tilde{A}\right)}$ is at most the clipping threshold.

### A.6 WEIGHT DECAY

The limit of decaying $w^l \leftarrow w^l(1 - lr \cdot wd)$ is just $A^l \leftarrow A^l(1 - lr \cdot wd)$.

### A.7 DECOUPLING LAYERS

We can actually use different $r$s and $M$s for every layer, and a similar limit theorem can be proved. We can also use different, independently sampled $\Omega$ for each layer ("untied $\Omega$s"). However, the limit would be exactly the same as before, as is apparent in our proof. In addition, we verify that even in finite $\pi$-nets, there is no difference in performance between tying $\Omega$ across layers or not (Fig. 5). All of our experiments are actually done with untied $\Omega$s.

## B EXPERIMENTAL DETAILS

All of our experiments are done on V100 GPUs. All of our networks use relu activation. The V-transform of relu is (Cho and Saul, 2009)

$$\mathcal{V}_{relu}(r_1 r_2 c, r_1^2, r_2^2) = \frac{1}{2\pi}\left(\sqrt{1 - c^2} + (\pi - \arccos(c))c\right) r_1 r_2$$

for any $r_1, r_2 > 0$ and $c \in [-1, 1]$.

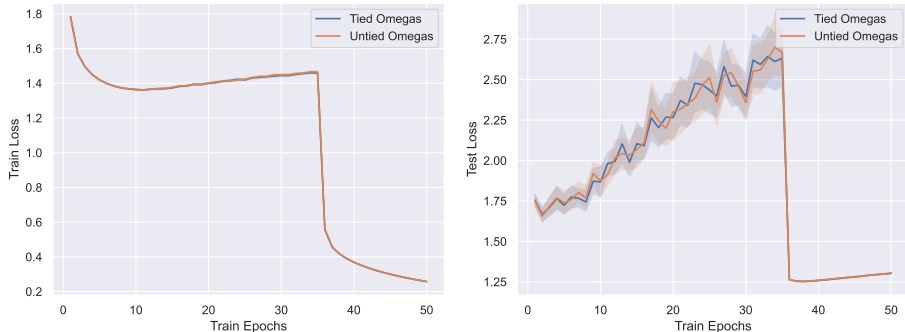

Figure 5: **Tying vs Untying $\Omega$ Across Layers Make No Difference in $\pi$-Nets.** Using the same procedure as in Section 4.1, we train $\pi$-net with $r = 400$ using the best hyperparameter combination we found (whose result shown in Table 1) 50 times, each with different independently sampled $\Omega$s, either with $\Omega$ tied across layers (blue curve) or not (orange curve). We plot their mean training loss and test loss curves here, with shade indicating 95% confidence interval.

## B.1 NETWORK INITIALIZATION AND PARAMETRIZATION

At initialization, 1) biases are always 0, 2) $M$ is set to $r$ so that $A^l, B^l$ are square matrices for $l = 2, \ldots, L$, and 3) we sample $A^l, B^l$ as standard Gaussians and then scale them as follows:

$$A^1 = A^1/A^1.norm(dim = 0, keepdim = True)$$
$$B^l = B^l/B^l.norm(dim = 1, keepdim = True) \quad \text{for all } l = 2, \ldots, L$$
$$A^l = A^l/\sqrt{A^1.shape[0]} \quad \text{for all } l = 2, \ldots, L, \text{ and}$$
$$A^{L+1} = 0$$

## B.2 FEATURE KERNEL

When we talk about the "feature kernel" of a $\pi$-limit $\mathring{f}^{\mathcal{P}}$, we always mean the $n \to \infty$ limit of the feature kernel of $f^{\mathcal{P}}$, and *not* the kernel induced by $g^L$ in $\mathring{f}^{\mathcal{P}}$. This feature kernel $K$ on inputs $\{\xi_1, \ldots, \xi_k\}$ is calculated as

$$K_{ij} = \mathcal{V}_\phi(\langle g_i^L, g_j^L \rangle, \|g_i^L\|^2, \|g_j^L\|^2)$$

where $g_i^L$ and $g_j^L$ are the $g^L$ in Theorem 3.4 evaluated for two inputs $\xi_i$ and $\xi_j$.

## B.3 CIFAR10

### B.3.1 $\mu$-NET, $\pi$-NET, AND $\pi$-LIMIT

For $\pi$- and $\mu$-networks, infinite or finite, we train for 50 epochs. We adopt a step learning rate schedule, with a learning rate drop of 0.15 at a milestone hyperparameter. We also clip gradients with a hyperparameter threshold, where the clip is triggered individually for each parameter by the parameter's norm, rather than total parameter norm.

**Hyperparameter Optimization** We first perform random search on hyperparameters listed in Table 2 for 2-hidden-layer $\pi$-net, $\pi$-limit, and $\mu$-net: We sample at least 512 hyperparameter combinations 1) for each $(width, r) \in \{128, 256, 512, 1024, 2048\} \times \{50, 100, 200, 400\}$ for $\pi$-net, 2) for each $width \in \{128, 256, 512, 1024, 2048\}$ for $\mu$-net, and 3) for each $r \in \{50, 100, 200, 400\}$ for $\pi$-limit. The best accuracies per $(width, r)$ are shown in Fig. 4(Middle).

Then we take the best hyperparameter combinations overall for $\pi$-net, $\pi$-limit, and $\mu$-net, and for $depth \in \{1, 2, 3, 4\}$ (where $depth$ denotes number of hidden layers), we resweep only learning rate and weight decay in a grid search, where the grid is $(lr^* \cdot \{2^{-2}, 2^{-1.5}, \ldots, 2^{1.5}\}) \times (wd^* \cdot \{2^{-2}, 2^{-1.5}, \ldots, 2^{1.5}\})$ and $lr^*$ and $wd^*$ are the optimal hyperparameters from the 2-hidden-layer sweep. The best test accuracies per depth are shown in Fig. 7, while the overall best test accuracies over all depth are shown in Table 1.

Table 2: CIFAR10 Hyperparameter Grid for $\mu$-Net, $\pi$-Net, and $\pi$-Limit

| Hyperparameter | Grid |
|---|---|
| Gradient Clip | $0.4 \cdot \{2^{-3}, 2^{-2} \ldots, 2^4\}$ |
| Learning Rate | $0.5 \cdot \{2^{-3}, 2^{-2} \ldots, 2^4\}$ |
| Weight Decay | $2 \cdot 10^{-5} \cdot \{2^{-3}, 2^{-2} \ldots, 2^4\}$ |
| Bias Mult | $0.5 \cdot \{2^{-3}, 2^{-2} \ldots, 2^4\}$ |
| LR Drop Milestone | $\{30, 35, 40\}$ |
| Input Weight LR Mult | $0.1 \cdot \{2^{-3}, 2^{-2} \ldots, 2^4\}$ |
| Output Weight LR Mult | $16 \cdot \{2^{-3}, 2^{-2} \ldots, 2^4\}$ |
| Input Weight Mult | $\{2^{-3}, 2^{-2} \ldots, 2^4\}$ |
| Output Weight Mult | $\{2^{-3}, 2^{-2} \ldots, 2^4\}$ |
| Batch Size | $\{4, 8, 16, 32\}$ |

Table 3: CIFAR10 Hyperparameter Grid for NNGP and NTK

| Hyperparameter | GP | NTK |
|---|---|---|
| Bias Variance | $\{2^{-4}, 2^{-3.5}, \ldots, 2^2\}$ | $0.5 \cdot \{2^{-4}, 2^{-3.5}, \ldots, 2^2\}$ |
| Bias LR Multiplier | n.a. | $\{2^{-4}, 2^{-3.5}, \ldots, 2^2\}$ |
| Input Weight LR Multiplier | n.a. | $0.5 \cdot \{2^{-4}, 2^{-3.5}, \ldots, 2^2\}$ |
| Output Weight LR Multiplier | n.a. | $\{1, 2^{0.25}, \ldots, 2^5\}$ |
| Ridge | $\{10^{-8}, 10^{-7} \cdots, 10^{-1}\}$ | |

### B.3.2 NNGP AND NTK

For NNGP and NTK, we perform kernel regression following Lee et al. (2018) using centered labels. For each $depth \in \{1, 2, 3, 4\}$, we sweep over hyperparameters listed in Table 3 (which change the kernels) using (complete) grid search, with weight initialization variance fixed at 1 in the NTK parametrization.[18] For each of NNGP and NTK, the best test accuracy over all depths is listed in Table 1.

### B.4 IMAGENET TRANSFER

We pretrain the $\pi$-Limit with $r = 200$ and 2 hidden layers for 30 epochs on a fixed subset of ImageNet32 with 250 (out of 1000) randomly subsampled classes. To evaluate on CIFAR10, we compute the kernel induced by the pretrained final-layer-features and perform kernel ridge regression. We optimize the hyperparameters in Table 4 via random search.

### B.5 OMNIGLOT

We focus on the 5-way, 1-shot task, with only 1 step of ANIL adaption.

### B.5.1 $\mu$-NET, $\pi$-NET, AND $\pi$-LIMIT

For $\pi$- and $\mu$-networks, infinite or finite, we train for 50 epochs, 1000 batches per epoch, and 8 tasks per batch. In each epoch, we validate on 500 batches from the validation set. Following Antoniou et al. (2019), we use cosine annealing learning rate schedule. We also clip gradients with a hyperparameter threshold, where the clip is triggered by the total parameter norm.

**Hyperparameter Optimization** We first perform random search on hyperparameters listed in Table 5 for 2-hidden-layer $\pi$-net, $\pi$-limit, and $\mu$-net: We sample at least 512 hyperparameter combinations 1) for each $(width, r) \in \{128, 256, 512, 1024, 2048\} \times \{50, 100, 200, 400\}$ for $\pi$-net, 2) for each $width \in \{128, 256, 512, 1024, 2048\}$ for $\mu$-net, and 3) for each $r \in \{50, 100, 200, 400\}$ for $\pi$-limit.

Then we take the best hyperparameter combinations overall (based on validation accuracy) for $\pi$-net, $\pi$-limit, and $\mu$-net, and for $depth \in \{1, 2, 3, 4\}$ (where $depth$ denotes number of hidden layers), we resweep only the meta learning rate (i.e. outer learning rate) and step size (i.e. inner learning rate) in a grid search, where the grid is $(ilr^* \cdot \{2^{-2}, 2^{-1.5}, \ldots, 2^{1.5}\}) \times (olr^* \cdot \{2^{-2}, 2^{-1.5}, \ldots, 2^{1.5}\})$

---

[18]This is without loss of generality because relu is homogeneous and we are sweeping the bias variance.

Table 4: ImageNet Transfer Hyperparameter Grid for $\mu$-Net, $\pi$-Net, and $\pi$-Limit

| Hyperparameter | $\pi$-Limit Transfer |
| --- | --- |
| Bias Mult | $0.5 \cdot \{2^{-3}, 2^{-2} \ldots, 2^3\}$ |
| Batch Size | $\{6, 8, 16\}$ |
| Learning Rate | $0.01 \cdot \{2^{-5}, 2^{-4} \ldots, 2^6\}$ |
| Input Weight Mult | $0.5 \cdot \{1.5^0, 1.5^{.25} \ldots, 1.5^{2.75}\}$ |
| Output Weight Mult | $\{2^{-0.5}, 2^0 \ldots, 2^3\}$ |
| Weight Decay | $\{2^{-7}, 2^{-6} \ldots, 2^0\}$ |
| Gradient Clip | $\{0.1, 0.2, 0.4, 0.6, 0.8, 0.9, 0\}$ |
| Ridge | $\{10^{-8}, 10^{-7} \cdots, 10^{-1}\}$ |

Table 5: Omniglot Hyperparameter Grid for $\mu$-Net, $\pi$-Net, and $\pi$-Limit

| Hyperparameter | Grid |
| --- | --- |
| Step Size | $0.5 \cdot \{2^{-2}, 2^{-1.75}, \ldots, 2^2\}$ |
| Meta Learning Rate | $16 \cdot \{2^{-3}, 2^{-2.75} \ldots, 2^3\}$ |
| Gradient Clip | $0.1 \cdot \{2^{-2}, 2^{-2.75} \ldots, 2^2\}$ |
| Bias Mult | $1 \cdot \{2^{-2}, 2^{-3.75} \ldots, 2^2\}$ |
| Input Weight Mult | $2 \cdot \{2^{-2}, 2^{-1.75}, \ldots, 2^2\}$ |
| Input Weight LR Mult | $0.2 \cdot \{2^{-2}, 2^{-1.75} \ldots, 2^2\}$ |

and $ilr^*$ and $olr^*$ are the optimal inner and outer learning rates from the 2-hidden-layer sweep. The best validation accuracies per depth are shown in Fig. 7. Then we take the models with overall best validation accuracies over all depth and evaluate them on the test set using 10000 batches. These test results are shown in Table 1.

To investigate the width non-monotonicity more thoroughly, we further perform random search on hyperparameters listed in Table 5 for 2-hidden-layer $\pi$-net and $\mu$-net. We sample at least 512 hyperparameter combinations 1) for each $(width, r) \in \{128, 256, 512, \ldots, 8192\} \times \{50, 100, 200, \ldots, 32000\}$ for $\pi$-net, and 2) for each $width \in \{128, 256, 512, \ldots, 8192\}$ for $\mu$-net. The best validation accuracies per $(width, r)$ are shown in Fig. 4(Right).

### B.5.2 NNGP AND NTK

As discussed in Section 4.3, for NTK and NNGP, ANIL meta-training has no effect, and meta-testing amounts to just taking 1 kernel gradient descent step on each task. We fix the step size (i.e. inner learning rate) at 0.5. For each $depth \in \{1, 2, 3, 4\}$, we sweep over hyperparameters listed in Table 6 using (complete) grid search, with weight initialization variance fixed at 1 in the NTK parametrization. For each of NNGP and NTK, the best test accuracy over all depths is listed in Table 1.

### B.5.3 VISUALIZATION OF IMAGE REPRESENTATIONS

We sample 5 random classes and 10 random images from each class from the Omniglot test set, for a total of 50 images. We take the best performing NTK, $\mu$-net, and $\pi$-limit and evaluate their feature kernels (c.f. Appendix B.2) on the 50 images. We then do PCA on these kernels to produce Fig. 1. In Fig. 9, we also do the same for our best performing $\pi$-net.

## C  DETAILED CALCULATIONS OF 1-STEP SGD

Suppose we present an input $\xi_0$ to the network and perform a step of gradient descent with learning rate $\eta$ and loss $\mathcal{L}$. Then simple calculations show that the updates $\Delta u, \Delta v$ to $u, v$ are

$$\sqrt{n}\Delta v = c\phi(\xi_0\sqrt{n}u), \quad \Delta u = cv \odot \phi'(\xi_0\sqrt{n}u) \tag{15}$$

where $c = -\eta\mathcal{L}'$. Then, via Tensor Programs, $f(\xi)$ for any $\xi$ now has a limit of the form

$$\lim_{n\to\infty} f(\xi) = \mathbb{E}(Z^{\sqrt{n}v} + Z^{\sqrt{n}\Delta v})\phi(Z^{\sqrt{n}u}\xi + Z^{\sqrt{n}\Delta u}\xi)$$

$$= \mathbb{E}(Z^{\sqrt{n}v} + \mathring{c}\phi(Z^{\sqrt{n}u}\xi_0))\phi(Z^{\sqrt{n}u}\xi + \mathring{c}Z^{\sqrt{n}v}\phi'(Z^{\sqrt{n}u}\xi_0)),$$

where $\mathring{c}$ is the deterministic limit of $c$ that is shown to exist by Tensor Programs.

Table 6: Omniglot Hyperparameter Grid for NNGP and NTK

| Hyperparameter | NNGP | NTK |
|---|---|---|
| Bias Variance | $0.1 \cdot \{2^2, 2^{2.5}, \ldots, 2^{10}\}$ | $\{2^{-1}, 2^{-0.5}, \ldots, 2^5\}$ |
| Bias LR Mult | n.a. | $\{2^{-4}, 2^{-3.5}, \ldots, 2^2\}$ |
| Input Layer LR Mult | n.a. | $0.1 \cdot \{2^{-4}, 2^{-3.5}, \ldots, 2^4\}$ |
| Output Layer LR Mult | n.a. | $0.1 \cdot \{2^{-4}, 2^{-3.5}, \ldots, 2^4\}$ |

Table 7: **Pretraining on ImageNet32 and Evaluating on CIFAR10, Full Results**. We pretrained $\mu$-net, $\pi$-net with $r = 200$, $\pi$-net with $r = 400$, and $\pi$-limit with $r = 400$ on ImageNet32 and evaluated the result model on CIFAR10. Here, the $\pi$-limit number 64.39 is the same as in Table 1 under $\pi$-*Limit ImageNet Transfer*. For reference, we also include the NNGP and NTK numbers in the left block. The * indicates we are comparing the $\pi$-limit transfer performance with $r = 200$ vs $\pi$-limit CIFAR10 number with $r = 400$, so the +2.79 is an *underestimate* of the improvement due to pretraining. The benefit of pretraining seems to be directly related to the capacity of the model, as $\pi$-Net with $r = 200 < \pi$-Net with $r = 400 < \mu$-Net $< \pi$-Limit.

| | NNGP | NTK | $\mu$-Net | $\pi$-Net $r=200$ | $\pi$-Net $r=400$ | $\pi$-Limit $r=200$ |
|---|---|---|---|---|---|---|
| Transfer | 58.92 | 59.63 | 61.84 | 58.02 | 59.36 | **64.39** |
| vs Table 1 | +0 | +0 | +0.53 | - | -1.28 | +2.79* |

## D    REMARKS ON THE $\pi$-LIMIT THEOREMS

1. $\pi$-projection actually ensures that, even for finite $n$, $f_t = f^{\mathcal{P}}$ for some $\mathcal{P}$. However, this $\mathcal{P}$ is random, with fluctuation coming from the sampling of $\Omega$ (which is fixed at initialization). Taking $n \to \infty$ reduces this fluctuation to 0.

2. Again, even with the projection, we are optimizing in an infinite-dimensional space, though "linearly infinite" $r \times \infty$ instead of "quadratically infinite" $\infty \times \infty$ like in the $\mu$-limit.

3. The forward pass of $\mathring{f}^{\mathcal{P}}$ can be thought of as that of another MLP with nonlinearity $\mathcal{V}_\phi$, as illustrated in Fig. 3. In this view, $M$ becomes the width of this MLP.

## E    NUMERICAL VERIFICATION OF THE $\pi$-LIMIT THEOREMS

Here we numerically show that wide $\pi$-nets have nearly identical loss curves as the $\pi$-limit. See Figs. 10, 11 and 13. In addition, Fig. 12 verifies the convergence of the feature kernel to its infinite-width limit.

## F    PROOFS

We will just prove Theorems 3.4 and 3.5, as Theorem 3.2 is a special case of them. At a high level, we need to do two things: 1) Show that $f_T$ converges almost surely to *something*, and 2) this *something* is $\mathring{f}^{\mathcal{P}_T}$. Here we assume the reader is familiar with Tensor Programs (Yang, 2019b;a; 2020a;b; Yang and Hu, 2020; Yang and Littwin, 2021; Yang et al., 2022), in particular the techniques used in Yang and Hu (2020).

### F.1    ALMOST SURE CONVERGENCE

Showing almost sure convergence is straightforward using the Tensor Programs technique, i.e. express $\pi$-initialization and the entire $\pi$-SGD training trajectory inside a Tensor Program (a NETSORT$^+$ program in particular) and apply the Master Theorem (c.f. Yang (2020b) in general and Yang and Hu (2020, Sec H.3, H.4) in particular). Below we sketch the program construction.

**Initial Vectors and Matrices**    Unlike the program for the $\mu$-limit, our program does not have initial matrix variables because in $\pi$-parametrization we do not initialize any $n \times n$ matrices as iid Gaussians. This means that we do not use `MatMul` instructions in our program. The initial vector variables in our program are just the $r$ columns $\Omega_{:1}, \ldots, \Omega_{:r}$ of $\Omega$.

Table 8: **Feature Kernel Regression (FKR) on CIFAR10**. We take the best performing $\mu$-net, $\pi$-net, and $\pi$-limit, and evaluate their learned feature kernels on CIFAR10 via kernel regression. We list their test accuracies in the middle block. For reference, we also include the NNGP and NTK numbers in the left block, as well as the ImageNet transfer results in the right block.

| | NNGP | NTK | $\mu$-Net | $\pi$-Net | $\pi$-Limit | $\pi$-Limit ImageNet Transfer |
|---|---|---|---|---|---|---|
| FKR | 58.92 | 59.63 | 59.12 | 59.72 | **61.85** | 64.39 |
| vs Table 1 | +0 | +0 | -2.19 | -0.92 | +0.35 | - |

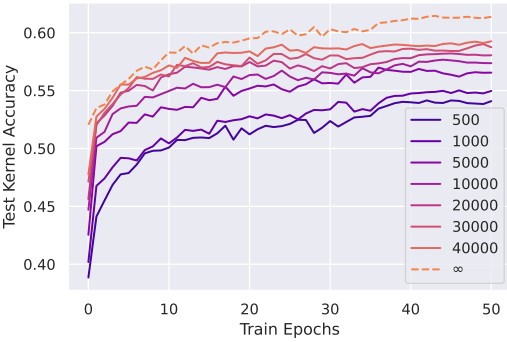

Figure 6: **Feature Kernel Regression of $\pi$-Nets vs Training Time, for Varying Widths.** We took our best performing $\pi$-limit and trained finite-width versions of it, for width from 500 to 40000. Throughout training, we measure their feature kernel regression accuracy. Altogether, we see consistent increase in performance across width at any moment in time. Note that the visible gap between $\pi$-limit and the widest $\pi$-net (even at initialization) is to a large extent due to the dependence of kernel regression accuracy on the smallest eigenvalues of the kernel. See Fig. 12.

**Network Preactivations**    All preactivations of the network will be of the form $\Omega C$ for some $C \in \mathbb{R}^r$ whose entries are scalar variables in the program, so that $\Omega C$ can be expressed as a vector variable using **Nonlin**$^+$.

**Weight Matrices**    We will sketch the constructions surrounding hidden weight matrices; the input and output weight matrices are similar and easier.

Like in Eq. (4), each hidden weight matrix in deep $\pi$-nets can be written mathematically as a sum of vector outer products.

$$w^l = \frac{1}{n} \sum_{s=1}^{M+t} (\Omega A_s^l) \otimes \phi(\Omega B_s^l), \quad \text{at time } t \tag{16}$$

where $A_s^l, B_s^l \in \mathbb{R}^r$ are the $s$th rows of $A^l$ and $B^l$. Here $\phi(\Omega B_s^l)$ is the activation going into $w^l$ at time $s$, and $\Omega A_s^l$ is the projected gradient $\Pi_\Omega \nabla_{h^l} \mathcal{L}$ at time $s$. Note the sum here is from 1 to $M + t$, where $M$ comes from the initialization and $t$ is from $t$ steps of training. All entries of $A^l$ and $B^l$ will be constructed as scalar variables in the program, so $\Omega A_s^l$ and $\phi(\Omega B_s^l)$ are both vector variables.

In the program, we do not express $w^l$ directly, but rather through its matrix-vector product with vectors such as the incoming activation $x^{l-1}$, which would be expressed via a combination of **Moment** and **Nonlin**$^+$ instructions like so:

$$w^l x^{l-1} = \sum_{s=1}^{M+t} \theta_s (\Omega A_s^l) \in \mathbb{R}^n \qquad \textbf{Nonlin}^+$$

$$\text{where} \quad \theta_s = \frac{1}{n} \langle \phi(\Omega B_s^l), x^{l-1} \rangle \in \mathbb{R} \qquad \textbf{Moment} \tag{17}$$

We express $(w^l)^\top$ indirectly through its matrix-vector products likewise, just with the roles of $(\Omega A_s^l), \phi(\Omega B_s^l)$ reversed.

**Gradient Projection**    In the program, we do not express $\Pi_\Omega$ directly, but rather indirectly by expressing the matrix-vector product $\Pi_\Omega v$ for vector variables $v$, such as $v = \nabla_{h^l} \mathcal{L}$. The projection

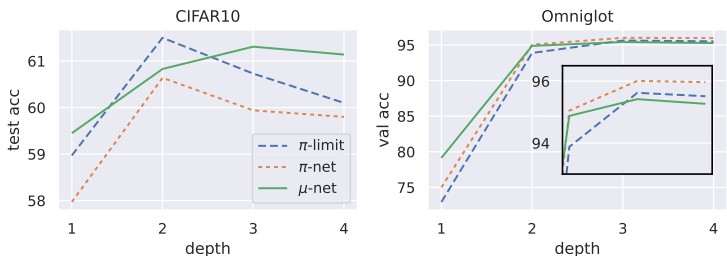

Figure 7: **Performance vs Depth on CIFAR10 and Omniglot.** We take best performing $\mu$-net, $\pi$-net, and $\pi$-limit from our thorough sweep of 2-hidden-layer networks, and resweep the learning rate and weight decay (for CIFAR10) or outer and inner learning rates (for Omniglot) for $\{1, 2, 3, 4\}$ hidden layers. We plot the best test accuracies of each depth here. See Appendix B for more details.

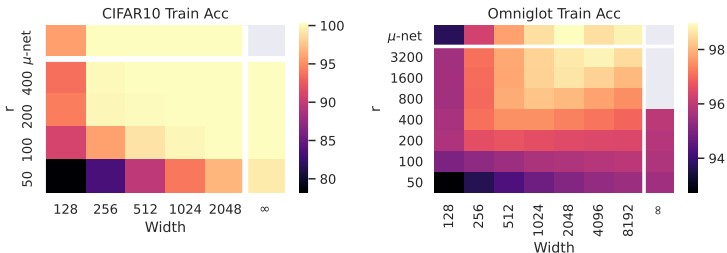

Figure 8: **Best Training Accuracy vs Width vs $r$ on CIFAR10 and Omniglot,** taken over all of our random hyperparameter searches. While networks with moderately large $r$ and width can overfit CIFAR10 completely, no $\mu$-net, $\pi$-limit, or $\pi$-net with width up to 8192 and $r$ up to 3200 is able to do so on Omniglot. See Appendix B.3.1 for experimental details.

matrix $\Pi_\Omega$ is mathematically equal to $\Omega(\Omega^\top\Omega)^+\Omega^\top$ (for any width), where $()^+$ denotes pseudo-inverse. In the program, we would first express (the entries of) $\frac{1}{n}\Omega^\top\Omega \in \mathbb{R}^{r\times r}$ using many **Moment** instructions

$$(\frac{1}{n}\Omega^\top\Omega)_{ij} = \frac{1}{n}\langle\Omega_{:i}, \Omega_{:j}\rangle \in \mathbb{R} \qquad\qquad \texttt{Moment}$$

Then its pseudo-inverse can be expressed as another **Moment** instruction (with purely scalar arguments).

$$(\frac{1}{n}\Omega^\top\Omega)_{ij}^+ = \frac{1}{n}\sum_{\alpha=1}^{n} f_{ij}(; \{(\frac{1}{n}\Omega^\top\Omega)_{ij}\}_{ij}) \in \mathbb{R} \qquad\qquad \texttt{Moment}$$

$$= f_{ij}(; \{(\frac{1}{n}\Omega^\top\Omega)_{ij}\}_{ij}) \in \mathbb{R} \qquad\qquad \texttt{Moment}$$

where $f_{ij}$ takes a matrix to the $ij$th entry of its pseudoinverse. Note the above expressions depend on $\Omega$ only, and not on $v$. Finally, we express $\Pi_\Omega v = \Omega\gamma \in \mathbb{R}^n, \gamma = (\frac{1}{n}\Omega^\top\Omega)^+\theta \in \mathbb{R}^r, \theta = \frac{1}{n}\Omega^\top v \in \mathbb{R}^r$ in the program like so

$$\Pi_\Omega v = \Omega\gamma = \sum_{i=1}^{r} \gamma_i\Omega_{:i} \in \mathbb{R}^n \qquad\qquad \texttt{Nonlin}^+ \qquad (18)$$

$$\text{where} \quad \gamma_i = \sum_{j=1}^{r}(\frac{1}{n}\Omega^\top\Omega)_{ij}^+\theta_j \in \mathbb{R} \qquad\qquad \texttt{Moment} \qquad (19)$$

$$\text{where} \quad \theta_j = \frac{1}{n}\langle\Omega_{:j}, v\rangle \in \mathbb{R} \qquad\qquad \texttt{Moment} \qquad (20)$$

**Wrapping Up** Other than what is discussed above, the unrolling of $\pi$-SGD follows identical to the unrolling of SGD in Yang and Hu (2020, Sec H.3, H.4). In particular, $f_t(\xi)$ for any input $\xi$ and time $t$ is a scalar variable in the program.

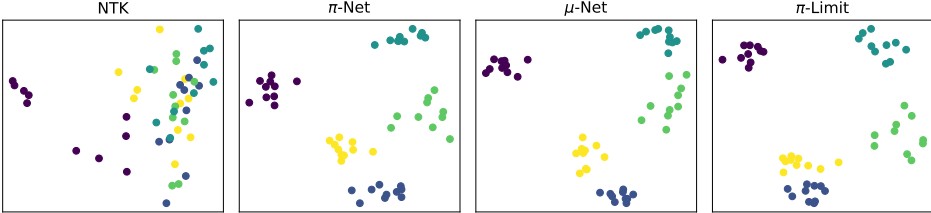

Figure 9: **PCA of representations of images from 5 classes in Omniglot test set.** Same setting as in Fig. 1, but here including our best performing $\pi$-net as well.

Table 9: **CIFAR10 Compute Time (in Seconds) Comparison**. We measure the average training time (in seconds) per epoch for 50 epochs of CIFAR10 using half precision on a NVIDIA V100 GPU. We evaluate a $\mu$-Net, $\pi$-Net, and the $\pi$-Limit, each of depth 1, 2, 3, and 4; the $\pi$-Nets and the $\pi$-Limits have $r = 400$. Because the $\pi$-Limit has a linearly increasing compute time per epoch, we also give an estimate expression for the compute time of the $\pi$-Limit in terms of $t$ epochs.

| Hidden Layers | $\mu$-Net | $\pi$-Net | $\pi$-Limit Average | $\pi$-Limit Epoch Estimate |
|---|---|---|---|---|
| 1 | 6.78 | 6.83 | 83.01 | $30.97 + 2.23t$ |
| 2 | 7.72 | 8.29 | 160.99 | $40.70 + 4.99t$ |
| 3 | 9.03 | 9.80 | 200.31 | $48.413 + 7.75t$ |
| 4 | 10.17 | 11.29 | 263.11 | $61.20 + 10.36t$ |

**Getting Almost Sure Convergence**  We apply the Master Theorem (Yang and Hu (2020, Thm 7.4) or Yang (2020b, Thm E.15)) to the program to get almost sure convergence to *some limit*. We just need to check the conditions of the theorem. They are all straightforward except that we need to check the pseudoinverse operation we took is almost surely continuous (in order to satisfy Yang and Hu (2020, Assm F.4(1))).[19] However, the only pseudoinverse we took was $(\frac{1}{n}\Omega^\top\Omega)^+$, and $\Omega$ has rank $r$ (full rank) almost surely for any $n > r$. Therefore, our pseudoinverse operation is almost surely continuous as pseudoinverse is continuous on matrices of constant rank.

### F.2  FORM OF THE LIMIT

**Forward Pass (Theorem 3.4)**  In the large-$n$ limit, by the Master Theorem, for hidden weights, Eq. (17) becomes

$$Z^{w^l x^{l-1}} = \sum_{s=1}^{M+t} \theta_s \sum_{i=1}^{r} A_{si}^l Z^{\Omega:i} = \sum_{i=1}^{r} Z^{\Omega:i} \sum_{s=1}^{M+t} \theta_s A_{si}^l$$

$$\text{where}\quad \theta_s = \mathbb{E}\left[\phi(Z^{h^{l-1}})\phi\left(\sum_{i=1}^{r} B_{si}^l Z^{\Omega:i}\right)\right]$$

Inductively, if $g^l \in \mathbb{R}^r$ represents the coefficients of $Z^{h^l} = Z^{w^l x^{l-1}}$ in terms of $Z^\Omega \stackrel{\text{def}}{=} (Z^{\Omega:1}, \ldots, Z^{\Omega:r})$ (which is distributed as a standard isotropic Gaussian vector), then

$$g_i^l = \sum_{s=1}^{M+t} A_{si}^l \mathbb{E}\,\phi(\langle g^{l-1}, Z^\Omega\rangle)\phi(\langle B_s^l, Z^\Omega\rangle),$$

which can be rearranged straightforwardly into the equation of Theorem 3.4. The equations for input and output layers can be derived similarly.

**Backward Pass (Theorem 3.5)**  For hidden weight $w^l$, since we maintain $w^l$ in the form of Eq. (16), the gradient update

$$w^l \leftarrow w^l - \eta\Pi_\Omega\nabla_{w^l}\mathcal{L} = w^l + (-\eta\Pi_\Omega\nabla_{h^l}\mathcal{L}) \otimes \phi(h^{l-1})$$

---

[19]Yang and Hu (2020, Assm F.4(1)) actually requires **Moment** nonlinearities with only parameter arguments to be continuous *eveywhere*, but because our result is almost sure anyway, we can ignore any measure zero event.

Table 10: **Omniglot Compute Time (in Seconds) Comparison**. We measure average training time (in seconds) per epoch for 50 epochs of Omniglot using half precision on a NVIDIA V100 GPU. We evaluate a $\mu$-Net, $\pi$-Net, and the $\pi$-Limit, each of depth 1, 2, 3, and 4; the $\pi$-Nets and the $\pi$-Limits have $r = 400$. Because the $\pi$-Limit has a linearly increasing compute time per epoch due to gradient concatenation, we also give an estimate expression for the compute time of the $\pi$-Limit in terms of $t$ epochs. This is not necessary for the 1-hidden-layer case, as there ANIL only trains the first layer, which does gradient accumulation.

| Hidden Layers | $\mu$-Net | $\pi$-Net | $\pi$-Limit Average | $\pi$-Limit Epoch Estimate |
|---|---|---|---|---|
| 1 | 28.68 | 29.33 | 36.48 | N.A. |
| 2 | 32.38 | 35.99 | 313.4 | $58.65 + 10.19t$ |
| 3 | 38.47 | 43.06 | 596.98 | $73.48 + 20.94t$ |
| 4 | 42.24 | 49.37 | 872.83 | $86.83 + 31.44t$ |

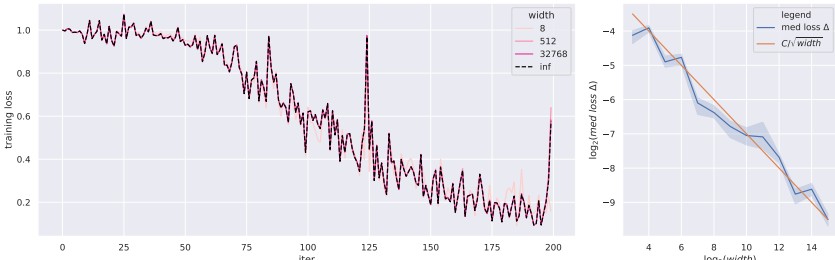

Figure 10: **Wide 1-hidden-layer $\pi$-nets with $r = 2$ have nearly identical loss curves as their $\pi$-limit. (Left)** We train $\pi$-nets of $r = 2$ and width $2^3, 2^9, 2^{15}$ as well as their common $\pi$-limit on a 128-image subset of CIFAR10 over 200 steps, with batch size 32 per step. We plot the training loss vs steps on the left. While the width-8 $\pi$-net fluctuates a bit around the $\pi$-limit curve, width-512 and -32768 $\pi$-nets have nearly identical loss curves as the $\pi$-limit. **(Right)** With the same dataset and training procedure, we sweep widths $2^{\{3,4,\dots,15\}}$ and 100 random seeds (which affect only the random initialization). For each width and seed, we calculate the median loss deviation of the $\pi$-net of that width from the $\pi$-limit, where the median is calculated over the 200 steps of training. Finally, for each width, we plot the median of those medians over the 100 seeds as the blue curve (with 95% confidence interval in shade). This curve shows that the loss curve of a $\pi$-net converges roughly as $1/\sqrt{width}$ to that of the $\pi$-limit.

in the limit corresponds to

$$A^l \leftarrow append(A^l, \text{coef}(Z^{-n\eta\Pi_\Omega\nabla_{h^l}\mathcal{L}})), \quad B^l \leftarrow append(B^l, \text{coef}(Z^{h^{l-1}}))$$

where $\text{coef}(Z^v)$ is the coefficient of $Z^v$ in terms of $Z^\Omega = (Z^{\Omega:1}, \dots, Z^{\Omega:r})$. (Note the factor of $n$ in $Z^{-n\eta\Pi_\Omega\nabla_{h^l}\mathcal{L}}$) is just there so that $n\eta\Pi_\Omega\nabla_{h^l}\mathcal{L}$ has $\Theta(1)$-sized coordinates). Of course, $\text{coef}(Z^{h^{l-1}})$ is just $g^{l-1} \in \mathbb{R}^r$ by definition. It remains to show that $\text{coef}(Z^{-n\eta\Pi_\Omega\nabla_{h^l}\mathcal{L}}) = \nabla_{g^l}\lim\mathcal{L}$, where $\lim\mathcal{L} = \lim_{n\to\infty}\mathcal{L}(f(\xi), y)$ is the loss at the limit (which is deterministic).

Using the Master Theorem, it is not hard to see that, for each preactivation $h^l$, $Z^{n\nabla_{h^l}\mathcal{L}}$ is the Frechet derivative $\frac{\partial\lim\mathcal{L}}{\partial Z^{h^l}}$ with respect to the random variable $Z^{h^l}$, where the Frechet derivative is defined with respect to the Hilbert space $\mathcal{H}$ of square integrable random variables in the $\sigma$-algebra generated by $Z^\Omega \overset{\text{def}}{=} (Z^{\Omega:1}, \dots, Z^{\Omega:r})$. Furthermore, using the limits of Eqs. (18) to (20) obtained from the Master Theorem, it is easy to see that

$$Z^{n\Pi_\Omega\nabla_{h^l}\mathcal{L}} = \Pi_{Z^\Omega}Z^{n\nabla_{h^l}\mathcal{L}}$$

where $\Pi_{Z^\Omega}$ is the orthogonal projection to the linear subspace of $\mathcal{H}$ spanned by the random variables $Z^{\Omega:1}, \dots, Z^{\Omega:r}$. Then Lemma F.1 applies and we get that

$$Z^{n\Pi_\Omega\nabla_{h^l}\mathcal{L}} = \sum_{i=1}^{r}\frac{\partial\lim\mathcal{L}}{\partial g_i^l}Z^{\Omega:i}$$

$$\text{coef}(Z^{n\Pi_\Omega\nabla_{h^l}\mathcal{L}}) = \nabla_{g^l}\lim\mathcal{L},$$

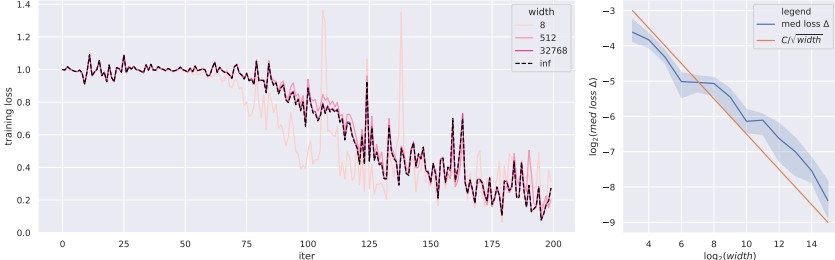

Figure 11: **Wide 2-hidden-layer $\pi$-nets with $r = 2$ have nearly identical loss curves as their $\pi$-limit.** We repeat the procedure in Fig. 10 for 2-hidden-layer $\pi$-nets. **(Left)** Width-32768 $\pi$-net has almost identical loss curve as the $\pi$-limit. Compared to the 1-hidden-layer case, the width 512 $\pi$-net is not as close to the limit, but this is expected as depth slows down convergence with width. **(Right)** Nevertheless, we still see $1/\sqrt{width}$ convergence to the $\pi$-limit in terms of training loss.

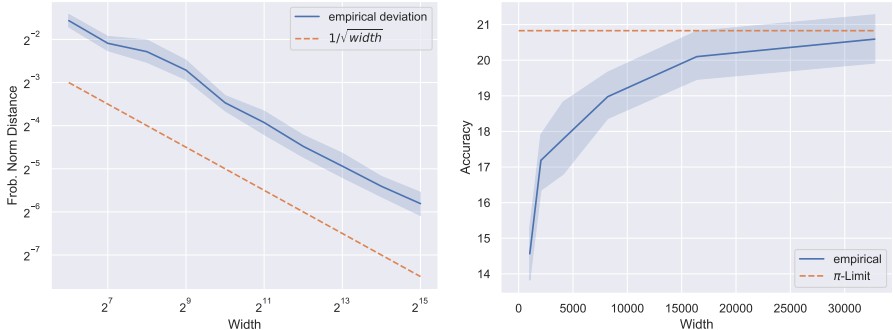

Figure 12: **Convergence of feature kernel at initialization, as measured by (left) Frobenius distance and (right) kernel regression accuracy.** We perform all experiments here on a subset of CIFAR10 with 400 training and 400 testing examples. **(Left)** We empirically verify that, at initialization, the feature kernels of $\pi$-nets (with 2 hidden layers, $r = 400$) converge to the feature kernel of the $\pi$-limit in Frobenius norm at a $1/\sqrt{width}$ rate. Here in blue we plot the Frobenius distance of the $\pi$-net feature kernel to the limit kernel, normalized by the Frobenius norm of the limit kernel. The shade represents 95% interval of the mean, taken over 10 random seeds. **(Right)** We compute the feature kernel regression accuracy of randomly initialized $\pi$-nets of different widths (blue solid curve) and their common $\pi$-limit (orange dashed curve). The shade represents 95% confidence interval of the mean, taken over 10 random seeds. We see a convergence of this accuracy as one would expect from the theory. However, note that because the stability of kernel regression depends crucially on small eigenvalues of the kernel, the width needs to quite large compared to the data size (= kernel size) in order to visibly see convergence of accuracy; for data size beyond 400 training samples, we cannot see such convergence for width $< 40,000$. This is why in Fig. 6, even at initialization we see a large gap in accuracy between $\pi$-nets and the $\pi$-limit.

where the derivative is now an ordinary partial derivative.

**Lemma F.1.** *Let $H$ be a Hilbert space and $V$ be a $k$-dimensional subspace of $V$, where $k$ is finite. Let $\mathcal{L} : H \to \mathbb{R}$ be a Frechet differentiable function. Suppose $\Gamma$ have for columns an orthonormal set of basis for $V$, and $w = \Gamma b \in V$ for some $b \in \mathbb{R}^k$. Then*

$$\Pi_V \nabla_w \mathcal{L}(w) = \nabla_b \mathcal{L}(\Gamma b),$$

*where $\Pi_V : H \to V$ is the orthogonal projection to $V$.*

*Proof.* By the theory of proximal gradients,

$$\Pi_V \nabla_w \mathcal{L}(w) = w - \min_{v \in V} \left[ \langle v, \nabla_w \mathcal{L}(w) \rangle + \frac{1}{2} \|v - w\|^2 \right].$$

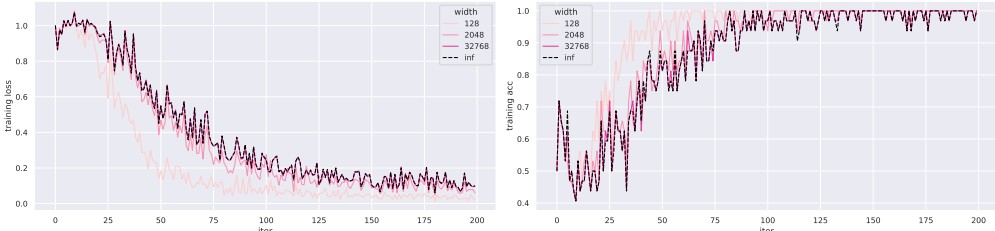

Figure 13: **Wide 2-hidden-layer $\pi$-nets with $r = 400$ have nearly identical loss curves as their $\pi$-limit.** We repeat the procedure in Fig. 10 for 2-hidden-layer $\pi$-nets, but now with $r = 400$ and widths 128, 2048, and 32768. **(Left)** Training loss. **(Right)** Training accuracy. In both subplots, width-32768 $\pi$-net has almost identical curves as the $\pi$-limit, and the width-2048 curves follow them closely.

Changing coordinates via $\Gamma$, we have

$$\Pi_V \nabla_w \mathcal{L}(w) = w - \min_{c \in \mathbb{R}^k} \left[ \langle \Gamma c, \Gamma \nabla_b \mathcal{L}(\Gamma b) \rangle + \frac{1}{2} \|\Gamma c - \Gamma b\|^2 \right]$$

$$= w - \min_{c \in \mathbb{R}^k} \left[ \langle c, \nabla_b \mathcal{L}(\Gamma b) \rangle + \frac{1}{2} \|c - b\|^2 \right]$$

$$= \nabla_b \mathcal{L}(\Gamma b)$$

$\square$

