# OpenReview forum: "Efficient Computation of Deep Nonlinear Infinite-Width Neural Networks that Learn Features"
_ICLR.cc/2022/Conference — ICLR 2022 Poster_

### Official Review · Reviewer_pNHj · 2021-10-29

**Correctness:** 3
**Technical Novelty And Significance:** 3
**Empirical Novelty And Significance:** 3
**Recommendation:** 6
**Confidence:** 3

**Main Review:**

The formal development presented in the paper is interesting and insightful. The idea is first presented in a simple example, in which the input and the output are a single real number, and this allows digesting the pretty complicated math. The idea itself, restricting the parameter dynamics in a few directions, is not new, as also stated by the authors  (it is also exploited in  Direct Feedback Alignment https://arxiv.org/abs/1609.01596, which in my opinion should be cited), but the explicit expression of the limit in the two theorems is, to the best of my knowledge, novel.

My main concern on the manuscript is on the meaningfulness of the numerical tests. The performance gap with respect to NTK on the CIFAR10 network is only 2 %, making the improvement brought by feature learning and by taking the explicit infinite-width limit difficult to appreciate. I suggest to repeat the test on a fully-connected architecture designed for image recognition, which achieves a test accuracy of 78 %, much closer to the state of the art on this data set: https://arxiv.org/abs/1511.02580

In this more realistic condition it should be possible appreciating better the improvement with respect to NTK and the nu-net.



**Summary Of The Paper:**

The paper introduces an approach (named pi-limit) to compute explicitly the infinite-width limit performance of a multilayer perceptron (MLP). The approach is based on projecting the gradient of the loss at each epoch in a fixed direction, chosen upon initialization. If one does so,  the expectation value of the function learned by the network at infinite training time can be computed explicitly by the iterative procedure described in Theorem 3.5. The asymptotic accuracy on a network trained on CIFAR10 is 61.5 %, slightly (0.2 %) better than the accuracy obtained by the nu-net procedure, which does not allow estimating explicitly the limit, and almost 2 % better than the the accuracy of a Neural Tangent Kernel, which, as well known, is feature agnostic. Based on these tests, it is concluded that the pi-limit is the first procedure which allows computing exact expectation values in a MLP and that, at the same time, allows learning features.

**Summary Of The Review:**

The theoretical developments presented in the paper are (to my knowledge) novel, and can trigger other ideas for further improvements. The numerical tests presented to support the usefulness of the theoretical framework are not fully convincing, but can be rather easily improved. My score would be a 7, if 7 was available.

---

> ### Author Response · Authors · 2021-11-10
> **Response**
>
> We appreciate the reviewer's time and feedback.
>
> > (it is also exploited in  Direct Feedback Alignment https://arxiv.org/abs/1609.01596, which in my opinion should be cited)
>
> Thanks for the suggestion. We will cite this work.
>
> > My main concern on the manuscript is on the meaningfulness of the numerical tests. The performance gap with respect to NTK on the CIFAR10 network is only 2 %, making the improvement brought by feature learning and by taking the explicit infinite-width limit difficult to appreciate.
>
>
> We disagree with the criticism that our numerical tests are not meaningful just because the performance gap on CIFAR10 seem "small", ignoring all of our other results on Imagenet transfer, Omniglot, and feature kernel regression.
>
> First of all, the CIFAR10 gap is not small.
> In fact, prior works such as [Lee et al. 2018](https://openreview.net/forum?id=B1EA-M-0Z) and [Lee et al. 2020](http://arxiv.org/abs/2007.15801) have claimed that kernel methods outperform MLPs based on the same margins (2%).
> In addition, the standard deviations of the pi-limit, pi-net, and mu-net accuracies on CIFAR10 due to initialization are all around 0.2%, so the 2% margin represents a 10-sigma event and thus is very statistically significant.
>
> Secondly, regardless of the final test accuracies, we clearly demonstrate that pi-net and pi-limit learn features on CIFAR10 through plots like fig 4(left) that show their feature kernel evolution.
>
> Thirdly, while we included CIFAR10 results as the community is most familiar with this dataset, other tasks like Omniglot or Imagenet transfer learning stress-test feature learning much more. Indeed, one of the most important ways feature learning is impactful in practice (and commercially) today is as part of the pretraining and finetuning paradigm (such as in BERT), and such tasks better reflect this procedures. Our results on them expose nontrivial performance gaps between NTK and the pi-limit (e.g., 40% gap on Omniglot), and are thus very meaningful in our opinion.
>
> Lastly, we have ongoing work on the pi-limit of more state-of-the-art architectures, which will amplify the gap with NTK, even on CIFAR10. In this work, we only wish to focus on the clean setting of MLP so the core insights of the pi-net and pi-limit are more apparent (and, e.g., not mixed up with possible confounders with architecture).
>
> > I suggest to repeat the test on a fully-connected architecture designed for image recognition, which achieves a test accuracy of 78 %, much closer to the state of the art on this data set: https://arxiv.org/abs/1511.02580
> > In this more realistic condition it should be possible appreciating better the improvement with respect to NTK and the nu-net.
>
> We appreciate the suggestion, but we have several reservations toward including it as an additional setting for CIFAR10: 1) the architecture is uncommon and would confound the insight obtained from such an experiment (likely raising concerns from other reviewers); 2) the proposed method involves unsupervised pretraining, which would be closer to our imagenet pretraining experiment (that already shows a nontrivial gap with NTK) than training on CIFAR10 from scratch.
>
> ---------
>
> If we have addressed the reviewer's concerns, please consider raising the score. Thanks again!

---

> > ### Comment · Reviewer_pNHj · 2021-11-22
> > **a comment on performance.**
> >
> > I understand that 2% of performance gap can be significant in the specific field of "exactly solvable" deep models.  However, quoting another referee "the performance is of course nowhere close to strong modern neural nets yet."  I believe that the experiment I suggested might help closing the gap, boosting enormously the importance of this work with no need of extending the framework (the architecture I suggest is fully connected, and, as correctly pointed out in your reply,  pretraining is algorithmically similar to transfer learning).

---

> > > ### Author Response · Authors · 2021-11-23
> > > **Thanks for the discussion!**
> > >
> > > While we agree with the general sentiment here and really appreciate the reviewer's ideas for improving our paper, our concerns about this specific paper still stands. To be more precise,
> > > - If the intention is reaching SOTA or modern networks, then this paper does not help because it is nowhere near 90+%.
> > > - If the intention here is to show that on the *vanilla* CIFAR10 setting there is a large gap, then it does not help either because it involves an uncommon pretraining + finetuning + conjugate gradient training procedure (see their section 4.4).
> > > - If being "vanilla" is not the point, then our Imagenet pretraining experiment already shows a large performance improvement over NTK.
> > > - If "CIFAR10" is not the point, then our Omniglot (which, in our opinion, is a *much much* better dataset to assess feature learning, and we hope the community will eventually agree) experiment already shows an enormous performance improvement over NTK.
> > >
> > > Now, we noticed that most of the improvement in the suggested paper comes from data augmentation (flips, rotation, shift). We are happy to explore this as an additional "nonvanilla" setting peer to our Imagenet pretraining experiment (without other gadgets from that paper to keep our experiment as clean as possible).

---

### Official Review · Reviewer_MBzR · 2021-10-31

**Correctness:** 3
**Technical Novelty And Significance:** 3
**Empirical Novelty And Significance:** 2
**Recommendation:** 6
**Confidence:** 3

**Main Review:**

Strengths:
Infinite-width models in their various flavours (NNGP, NTK, ...) form a useful set of tools for theoretically understanding the behaviour of neural networks in terms of classical kernel methods or GPs. Most (all?) of these types of limits usually mean that the limiting model keeps some notion of feature space constant during training. Motivated by the fact that practical neural networks actually update the feature space during training, the authors introduce a new limiting model (the pi-limit) that allows the features of the kernel to be updated during training. This is an important direction for the progression of infinite-width models.

Main criticism:
As pointed out by the authors, there is a gap between these models and modern neural networks in terms of the architectures, training schemes, and training regime (i.e. is the network "wide enough" and trained with a "small enough" learning rate?). This means that when performing this style of analysis, one usually has to make some modification to what is typically done in practice. One runs the risk of making a modification so severe that the analysed model does not reflect a neural network that people usually compute with.

In this work, in order for the infinite width model to be tractable, several modifications are required for 1 layer case:
-Initialise readout parameters at 0
-Projected gradients (but only for hidden layers)
-Minibatch size 1 (?)
-Gradients are appended to the network instead of added (!)

In the multi-layer case, an additional modification is required:
-pi-Initialisation. This initialisation appears to be substantially different to the typical scaling by sqrt n that is typical in infinite width works and in vanilla finite-width initialisation schemes (e.g. Glorot init).

These modifications, especially the fact that gradients are appended instead of added/subtracted in the usual way gradient updates are applied, mean that the model studied is quite different to one that would typically be used. This makes me wonder whether the studied model bears any resemblance to "real" neural networks. The model also does not achieve state of the art (it is not capable of expressing convolutional or transformer architectures). Since the model is not reflective of any simple practical neural network, nor does it achieve good performance, I am questioning the interest of this model in the broader ML community.

Questions for the authors:
1. Can you say a bit more about this? "For simplicity, we only consider batch size 1; it’s straightforward to generalize to larger batch sizes". There is a comment in appendix A (A.4), but I do not see any statement or how to modify the result toa accommodate larger batch sizes.

2. In equation (6), we use the V-transform, which is essentially the neural network kernel. The neural network kernel is symmetric in its second and third argument (i.e. swap X and Y and you get the same thing). In equation (6), the second and third argument don't look similar at all. Can you explain this asymmetry?

3. There are number of fiddly modifications performed in the experimental evaluation. For example, the learning rate drop mentioned in footnote 13, gradient clipping, ANIL, cosine annealing, ... . This seems slightly counter to the overall objective of the paper, which was (I think) primarily to study an infinite width network that is able to perform feature learning for its theoretical niceness. When moving to the empirical evaluation, I am not convinced that the same amount of hyperparameter and model tuning was afforded to the NTK/NNGP/finite width models as was given to the pi-limit. I think that the fairest comparison would be to not include any "tricks" in any models.

**Summary Of The Paper:**

The authors study a certain variant of an MLP trained using a projected gradient descent-inspired update rule in the infinite width limit. This infinite width model admits closed-form rules for computing the predictor at any training iteration (as long as the V-transform or neural network kernel being known, which is the case for e.g. the arc-cosine kernel). The authors evaluate their model on Omiglot and CIFAR10 against NTK and finite width baselines.

**Summary Of The Review:**

After the reviewer response, I have upgraded my score to 6. My main concern remains that the model analysed is different to what is typically used in practice.

The paper appears to be technically sound, novel and written with the intention of solving an important problem. But it studies a model so far removed from a typical neural network (so it's theory is not widely applicable), and the model does not achieve state of the art performance (so it is empirically not widely applicable).

---

> ### Author Response · Authors · 2021-11-09
> **(1) There seems to be some important misunderstandings**
>
> Thanks Reviewer MBzR for your time.
>
> > Main criticism: As pointed out by the authors, there is a gap between these models and modern neural networks in terms of the architectures, training schemes, and training regime (i.e. is the network "wide enough" and trained with a "small enough" learning rate?). This means that when performing this style of analysis, one usually has to make some modification to what is typically done in practice. One runs the risk of making a modification so severe that the analysed model does not reflect a neural network that people usually compute with.
>
> > In this work, in order for the infinite width model to be tractable, several modifications are required for 1 layer case: -Initialise readout parameters at 0 -Projected gradients (but only for hidden layers) -Minibatch size 1 (?) -Gradients are appended to the network instead of added (!)
>
> > In the multi-layer case, an additional modification is required: -pi-Initialisation. This initialisation appears to be substantially different to the typical scaling by sqrt n that is typical in infinite width works and in vanilla finite-width initialisation schemes (e.g. Glorot init).
>
> > These modifications, especially the fact that gradients are appended instead of added/subtracted in the usual way gradient updates are applied, mean that the model studied is quite different to one that would typically be used. This makes me wonder whether the studied model bears any resemblance to "real" neural networks.
>
> There seems to be some important misunderstanding regarding our modifications. Our main modifications for a finite network are just 1) gradient projection, and 2) pi-initialization. We call such modified networks *pi-nets*. In finite-width pi-nets, gradients are *added* to the weights (after projection), as one would usually do. Mathematically, in the infinite width limit, this translates to a sort of gradient appending --- this is not of our design, but rather just a mathematical consequence of taking the infinite-width limit of projected gradient descent. Criticizing that the pi-limit does gradient appending instead of gradient accumulation would be like criticizing kernel regression with NTK is not an SGD algorithm, so is not relevant to the study of neural networks.
>
> In the 1-hidden-layer case, zero readout init and minibatch size 1 are just pedagogical simplifications to make the math more clear to first time readers. Our main experiments use larger batch size up to 32 (see table 2 and 4). So the proper comparison is of our (pi-init, gradient projection) against (standard init, SGD with no projection). As shown in table 1, in finite networks, the difference in performance between a pi-net and a mu-net (or a standard network) is small, especially compared to the difference to NTK. In addition, as seen in e.g. fig 8, pi-net exhibits the same feature learning behavior as mu-net. Thus, we believe pi-nets serve as a good model for studying feature learning, and our modifications are not consequential in that regard.
>
> Please let us know if our response here clears up your confusion, as we believe this is an important point.
>
> We will respond to your other points in additional comments.

---

> > ### Comment · Reviewer_MBzR · 2021-11-10
> > **I was wrong**
> >
> > I mistakenly thought that the appending of gradients was due to construction, but in fact this property emerges from the projected gradient descent setup.

---

> > > ### Author Response · Authors · 2021-11-17
> > > **Thanks!**
> > >
> > > We are happy that the rebuttal has allowed us to come to an agreement. Please consider raising your score if we have addressed your concerns. Thanks!

---

> ### Author Response · Authors · 2021-11-09
> **(2) Additional Responses**
>
> > The model also does not achieve state of the art (it is not capable of expressing convolutional or transformer architectures). Since the model is not reflective of any simple practical neural network, nor does it achieve good performance, I am questioning the interest of this model in the broader ML community.
>
> Hopefully our previous response convinces you that pi-net *is* reflective of standard MLPs in regards to performance and feature learning.
>
> Our goal with the empirical results is to further show that the pi *limit* is reflective of standard MLPs in regard to performance and feature learning (see fig 1 and table 1), and hence this limit may be a valuable idealization to study neural networks and of interest to the broader ML community. Achieving SOTA has never been the goal here. (But note, actually our Omniglot results are state-of-the-art for MLPs (matching the results achieved in Finn et al. which had to use Adam and Batchnorm), but we do not highlight this in our paper because it's besides our point)
>
> It’s actually straightforward to generalize pi-net to convolutions and transformer architectures, and we have ongoing work to this end. However, this submission is intended to focus on the simple case of MLP to cleanly study the mathematics involved in and basic observations of the pi-limit, without delving into the complexities and the engineering details entailed by SOTA architectures, which we believe would have confused most readers.
>
> > Can you say a bit more about this? "For simplicity, we only consider batch size 1; it’s straightforward to generalize to larger batch sizes". There is a comment in appendix A (A.4), but I do not see any statement or how to modify the result to accommodate larger batch sizes.
>
> To accommodate a batch of size M, one can just compute M forward/backward passes, one for each element of the batch, and, for each A^l, obtain M vectors that would be appended to A^l. One then just appends those vectors to A^l, increasing its number of rows by M. Likewise for B^l. The only exception is in the first layer, where we still accumulate the gradients onto A^l. In practice, we can naturally batch the M forward/backward passes instead of doing them one-by-one, and this is how we implemented the calculation.
>
> We will expand on section A.4 with this explanation.
>
> > In equation (6), we use the V-transform, which is essentially the neural network kernel. The neural network kernel is symmetric in its second and third argument (i.e. swap X and Y and you get the same thing). In equation (6), the second and third argument don't look similar at all. Can you explain this asymmetry?
>
> We are not exactly sure we understand the confusion here. In eq (6), we can swap the 2nd and 3rd arguments without changing the mathematics, as you correctly remarked, but that doesn't imply that the 2nd and 3rd argument should "look similar".
>
> The 2nd and 3rd arguments describe the “variances” of the Gaussian variables involved in the V-transform expectation. In our case, one of the variances, the B_t o B_t term, comes from the 2nd layer weights, while the other variance, C_t^2 \xi^2 1, comes from the first layer preactivation; the V-transform itself describes the limit of the contraction between the (B part of) 2nd layer weights and the activation of the first layer. Weights and activations are naturally asymmetric, hence why eq (6) is likewise asymmetric.

---

> ### Author Response · Authors · 2021-11-09
> **(3) Additional Responses**
>
>
> > There are number of fiddly modifications performed in the experimental evaluation. For example, the learning rate drop mentioned in footnote 13, gradient clipping, ANIL, cosine annealing, ... . This seems slightly counter to the overall objective of the paper, which was (I think) primarily to study an infinite width network that is able to perform feature learning for its theoretical niceness. When moving to the empirical evaluation, I am not convinced that the same amount of hyperparameter and model tuning was afforded to the NTK/NNGP/finite width models as was given to the pi-limit. I think that the fairest comparison would be to not include any "tricks" in any models.
>
> We disagree. The realistic training procedures (e.g., ANIL, cosine annealing) and the extensive hyperparameter coverage are features, not bugs, of our work. Prior works such as [Lee et al. 2018](https://openreview.net/forum?id=B1EA-M-0Z) or [Lee et al. 2019](http://arxiv.org/abs/1902.06720) came to misleading conclusions (e.g. that NNGP of MLP is better than finite-width MLP, or NTK accurately reflects finite neural networks) due to the restricted settings they worked in (e.g. no learning rate drop, or very small learning rates) that are not reflective of how neural networks are used in practice. Had they used a more realistic training procedure or more hyperparameter optimization, they would have found the opposite conclusion. We seek to avoid such mistakes by focusing on realistic scenarios.
>
> For example, ANIL and cosine annealing are fairly standard in metalearning. The learning rate drop is much closer to what people do in practice on datasets like CIFAR10 and Imagenet. Like we remark in footnote 13, the kernel methods would obtain the same results if translated to kernel gradient descent for infinite-time with a single learning rate drop.
> Gradient clipping is there to compensate for the fact that typically metalearning would train using Adam, but we are limited to SGD in this work. To be consistent, we use it for CIFAR10 as well.
> All of these modifications were done for finite width models, so our comparison with them is fair.  For NTK/NNGP, we also implemented these modifications where they have any effect. For example, we evaluated them using ANIL as well on Omniglot. But on CIFAR10, as remarked above, learning rate drop and gradient clipping would have no effect on the kernel regression, so it did not make sense to sweep these hyperparameters. On Omniglot, ANIL metatraining has no effect on kernel methods; as remarked in footnote 17, we tried 2nd order MAML as well but kernel performance strictly decreases with metatraining (even with gradient clipping) because the randomization of labels across tasks confuse the readout layer. Therefore, for our main results, it did not make sense to implement cosine annealing and gradient clipping --- as they have no effect --- for kernel methods on Omniglot.
>
> As suggested by reviewer emnb, we are happy to include in the appendix an additional suite of results with just simple reasonable hyperparameters if this alleviates the reviewer's concerns.

---

### Official Review · Reviewer_JW6k · 2021-11-01

**Correctness:** 4
**Technical Novelty And Significance:** 4
**Empirical Novelty And Significance:** 4
**Recommendation:** 8
**Confidence:** 3

**Main Review:**

Overall, this paper reads well and is clear despite the dense mathematical content, and solves a relevant problem.
The solution is technically sound, and works well empirically.
I would be very happy to see this paper accepted.

I still have some minor comments and concerns, however, that I hope the authors would consider addressing:

- The proposed model ends up being quite different to a standard MLP, both in the way the weights are computed and the way training is performed.
This is unlike the previously proposed $\mu$-net, which is much closer to the way standard MLPs are trained.
Clearly, the choices made are necessary for the computational speedups that this paper is about.
However, I was hoping that the authors could comment more on the intuitive differences between these models?
The authors briefly discuss a similarity to the hypernetworks from Ha et al. (2016), but it would be great if this discussion could be expanded, particularly with respect to $r$.
For instance, would the authors expect the inductive biases to be much different between the two models?
This is likely a difficult question to answer, so I would simply ask the authors for some additional discussion along these lines.
However, this question is quite relevant for determining what we can learn from these infinite width limits, which is one of the main motivations of studying infinite width limits.

- I think that the results from Table 7 in the Appendix should be promoted to the main text.
While not explicitly stated, I believe I am correct that the training loss in Table 1 is the standard softmax for the feature learning methods?
In this case, to ensure a fair comparison between methods, it is important that the same loss be used, as in my experience FKR is very different to training with a proper classification loss.
Along these lines, do the authors have any intuition as to why the $\mu$-net suffers much more than the $\pi$-net when used for FKR?

- Would the authors be able to provide a comparison to $\mu$-net and $\pi$-net under transfer learning?
It would be understandable if not due to the computational requirements of doing so, but I think it would strengthen the paper.

**Summary Of The Paper:**

In this work, the authors propose a new feature learning limit for infinite-width neural networks that resolves some of the computational issues with the previous $\mu$-limit.
This new limit, called the $\pi$-limit, is based on projected gradient descent, and crucially allows the authors to evaluate the model on deep networks, which could not be done before.
The authors show that it outperforms both finite-width and previous infinite-width networks on CIFAR10 and Omniglot.

**Summary Of The Review:**

In summary, I think this is a very good paper that I would personally like to see at the conference.
I have some minor reservations which I hope the authors might address, but unless the other reviewers find something critical that I have missed I am unlikely to change my score.
Nevertheless, I look forward to reading the other reviewers' thoughts and the authors' responses.

---

> ### Author Response · Authors · 2021-11-14
> **Response**
>
> We appreciate the reviewer's time and feedback.
>
> > However, I was hoping that the authors could comment more on the intuitive differences between these models?
>
> Definitely, at initialization, we expect pi-limit's features to be inferior to those the mu-limit because of the low rank structure of the former; this is confirmed by Fig 4 (left). However, over the course of training, the pi-limit seems to learn very similar features as a mu-net, as shown in fig 1 and 8, and we expect they are similar to the learned features of a mu-limit as well, if we can compute it out. So our intuition currently is that, even though pi-net/pi-limit and mu-net/mu-limit start as very different models at initialization, after training for long enough, they capture very similar features and reach similar performance. For example, currently we have no reason to suspect that, if we are able to train BERT's pi-limit, we would get significantly worse language features. But of course, this expectation can change in the future as more evidence arises.
>
> > More discussion on connection with hypernetworks, especially in relation to $r$.
>
> Eq (10) shows that the weights $w^l$ in the middle layers of a pi-net can be thought of as generated by feeding in the random $\Omega$ into 1-hidden-layer MLP $F_{A^l, B^l}$ with weights $A^l, B^l$ and computing
> $$w^l_{\alpha\beta} \gets \frac 1 n \langle \Omega_\alpha, F_{A^l, B^l}(\Omega_\beta) \rangle.$$
> This MLP $F_{A^l, B^l}$ has input dimension $r$, hidden dimension $M$ (which is the dimension that grows as the pi-limit is trained), and output dimension $r$.
>
> This generation scheme in fact is not specific to just initialization: By Thm 3.5, at any time $t$, the pi-limit has the form $\mathring f^{\mathcal P_t}$, and we can *generate* a finite-width pi-net from the pi-limit's $\mathcal P_t$ by sampling a fresh $\Omega$ and feeding it into the above scheme. If the finite width is large enough, then we expect this generated pi-net to approximate the pi-limit well.
>
> So one may compare this pi-net generation procedure to how a GAN (pi-limit, or its $\mathcal P_t$) generates an image (pi-net) from random noise ($\Omega \in \mathbb R^{n\times r}$, which can be thought of as a batch of $n$ vectors of dimension $r$). However, this comparison does not capture the difference in the behavior of the "batch dimension" n. While a GAN would generate n images given n noise samples, a pi-limit would generate a *single* width-n network from n noise samples.
>
> Likewise, despite the superficial similarity that both pi-net and hypernetworks have some sort of "weight generator", there are several important differences:
> - pi-net generation procedure involves random noise $\Omega$, while hypernetworks generation is deterministic.
> - pi-net weight generation is input agnostic, while hypernetworks generation of a layer's weight matrix depends on the activation feeding into that layer.
> - the same generator can generate pi-net of any width, but a hypernetwork's weight dimensions are fixed before training.
>
> In summary, while GAN or hypernetworks help us intuit different aspect of pi-net/pi-limit by analogy, they remain completely distinct models.
>
> > While not explicitly stated, I believe I am correct that the training loss in Table 1 is the standard softmax for the feature learning methods?
>
> Yes.
>
> > I think that the results from Table 7 in the Appendix should be promoted to the main text.
>
> Sure. If we have enough space for camera-ready, we will do so.
>
> > Along these lines, do the authors have any intuition as to why the -net suffers much more than the -net when used for FKR?
>
> In that table, we just did FKR on the best performing models in terms of test accuracy, without optimizing for FKR accuracy directly. If we directly optimize FKR accuracy, mu-net might not suffer as much as pi-net. In addition, the finite network FKR accuracies can have some variance to them from initialization, so it's possible on a different initialization, the mu-net's performance penalty would not be so severe.
>
> > Would the authors be able to provide a comparison to -net and -net under transfer learning?
>
> Yes we should have these results by camera-ready, if our work is accepted.
>
> ----------
>
> Thanks again for your feedback. Let us know if you have any other questions :)

---

> > ### Comment · Reviewer_JW6k · 2021-11-25
> > **Thanks for your response!**
> >
> > Thanks to the authors for their response, as well as to the other reviewers for their detailed reviews - I am admittedly not really an expert in this area and so these were very helpful to read. Overall, I am not slightly less enthusiastic about this work than I was when I initially read it. I think there are a few items that the authors should address in future revisions, although I am not expecting the authors to address this before the discussion period ends.
> >
> > In my mind, the key issue that the authors should spend more time addressing is whether the $\pi$-net and $\pi$-limit really tells us much about what happens in standard, finite width networks. This seems to be a concern echoed by the other reviewers. In particular, I find myself agreeing with reviewer emnb, who notes that since the $\pi$-limit isn’t very practical, and creates further distance between itself and standard practice for training finite-width neural networks, it is unclear what is to be gained. Reviewer emnb has also expressed concerns about the gap between (practical) $\pi$-nets and the $\pi$-limit; while their concerns are important if true, I am admittedly not entirely convinced of their arguments here.
> >
> > Reviewer emnb has also raised an issue with the claim that $\mu$-net result in Table 1 will be representative of the best standard parameterized neural network in practice. Looking at the arguments further, I am somewhat inclined to believe this. Would the authors be able to settle this empirically by doing a hyperparameter search over standard networks? Hopefully this shouldn’t be too computationally intensive for CIFAR-10.
> >
> > Finally, I would like to note that I am not convinced by the arguments that some reviewers have made surrounding the outright performance of these methods, and how close they get to SOTA.
> >
> > In summary, I am less in favor of this paper now than I was on my initial reading. Ideally I would give this paper a score of 7; however as this is not an option I am willing to give the authors the benefit of the doubt and leave my score at 8.

---

> > > ### Author Response · Authors · 2021-11-30
> > > **Thanks for the update!**
> > >
> > > We will do another sweep in standard parametrization as you suggest, as well as address what "the pi-net and pi-limit really tells us much about what happens in standard, finite width networks."

---

### Official Review · Reviewer_emnb · 2021-11-02

**Correctness:** 2
**Technical Novelty And Significance:** 4
**Empirical Novelty And Significance:** 3
**Recommendation:** 6
**Confidence:** 4

**Main Review:**

Thanks a lot for the replies to many questions I asked!

Through the rebuttal, some central claims and contributions are clearer to me. But truthfully it turns out the paper now appears less promising and less exciting than I originally thought. Two major points that I learned from the rebuttal:

- The $\pi$-limit appears to give a qualitative view of the $\pi$-net, but not the more precise details of the $\pi$-net unless the width is 10 or 100 times larger than $r$. For example, the second figure in figure 12 shows that at width 2048 or even 32786, for $r=400$, the training accuracy could deviate from the $\pi$-limit as much as 10% at various points of time. Figures 9 and 10 show better matches, but it is for $r=2$ which is atypical and very low.

- The $\pi$-limit can take a very long time to compute; its runtime is multiple times longer than training a $\pi$-net (Table 8). Rather than “efficient computation”, what is demonstrated here is basically computability, against the fact the $\mu$-limit is practically uncomputable.

Without further demonstration of what use the $\pi$-limit can offer, this is quite hard to appreciate the $\pi$-limit, against the empirical performances which not all reviewers here are excited about. Note, throughout the infinite-width literature, be it NTK or mean field, it is quite standard that the identification of the infinite-width limit comes with an additional analysis or a demonstration of use, such as proving theoretical global convergences.

I’m also empathetic with reviewer MBzR’s concern, which is that the $\pi$-parameterization simply goes further away from the usual initialization practice. The fact that the $\mu$-parameterization bears similarity to the usual initialization practice (even though strictly speaking, it actually differs from the practice in a very important way) is probably one major reason why people pay much attention to it. By differing even more, the $\pi$-limit disregards entirely this premise. As such, I would expect to see from the $\pi$-limit something more interesting than a “fix” to the $\mu$-limit.

In various ways, I currently find less reasons to be convinced it’s a promising direction.

I will give the benefit of the doubt and raise my score, not because I want to align with other reviewers, but mainly because I simply like the idea. But I wish the paper were more carefully fleshed out than just this.

=================================================

The proposal is interesting and novel to my knowledge. The fact that it allows for computation of the infinite-width limit, unlike $\mu$-parameterization, while still achieving some nontrivial performance, is valuable in my opinion. I find that very encouraging, despite that the performance is of course nowhere close to strong modern neural nets yet. I also like that the paper provides a short description of the difficulty with $\mu$-limit and takes a detour to the 1-hidden-layer example.

I however find that the paper incomplete in several crucial aspects. The following should be considered in the finalized version:

- The paper does not demonstrate convincingly if a finite-width net under $\pi$-parameterization approaches the $\pi$-limit as widths go to infinity; that is, a verification of Theorem 3.5. Now that the limit can be computed (as compared to $\mu$-limit), this should have been done. In particular, it should demonstrate the closing gap to the $\pi$-limit at every training steps, as indicated by Theorem 3.5. How large should the width be to be visually close to the limit? Is it close in just the classification accuracy metric? These questions should have been the first items to be answered when an infinite-width limit is claimed to appear more practical.

    Figures 4 and 5 are insufficient in this aspect. Figure 5 looks concerning: as it shows, a width of 40,000 (!) still has a big gap to the limit on CIFAR10. I think this warrants deeper investigation.

- The hyperparameter $r$ appears very important, but there is little discussion. From the experiments, $r$ should be large. But the need for large $r$ complicates the studies.

    - Firstly this questions the relevance of the $\pi$-limit itself, if one takes it as a theoretical limit of $\pi$-net. In particular, $\pi$-limit is derived with $\text{width}\to\infty$ while keeping $r$ constant. Once $r$ is a nontrivial fraction of the width, one should expect some nontrivial random matrix effect to kick in and the $\pi$-nets to have nontrivial deviation from the $\pi$-limit, casting doubt on the validity of $\pi$-limit. This is indeed the case in the experiments.

    - Secondly if the proposal is to consider $\pi$-limit as a new neural architecture with a specific training algorithm rather than a theoretical infinite-width limit of $\pi$-nets, then the comparison should have been done differently. That is, it should not be compared against the simple NTK of feedforward nets, its expressiveness property should be investigated, its runtime should be studied, etc.

- The claim (in the paper title) is that $\pi$-limit can be efficiently computed, but again there is little discussion. In particular, can the paper provide a snapshot of the runtime for a reasonable, easily reproducible setup? There is a very short remark in page 4 that the complexity scales as $O(T^2)$, but this is insufficient to claim efficient computation. Is it running faster than or comparably to SGD on $\pi$-nets, in terms of compute hours? Is it better / faster to just run SGD on $\pi$-nets, and for what range of $r$?

Other issues:

- The paper claims outperformance of $\pi$-limit and $\pi$-nets over others in Table 1, but some of these numbers look too close to make a call. An increase of 2% for CIFAR10 could be borderline (given the nontrivial randomness in the generation of the matrices $A_l$ and $B_l$).

- Footnote 12 is quite concerning: it is well-known that standard-parameterized MLPs should behave differently from $\pi$-limit (and are close to NTK at best) as widths tend to infinity, but this footnote seems to suggest that the experimented $\pi$-nets may operate in a regime far from the $\pi$-limit?!

- Footnote 13 seems unfair. Firstly the design of the learning rate schedule should be considered as a knob in hyperparameter tuning. Secondly the outcome of convex optimization would differ if one switches from a constant learning rate schedule to one with learning rate drops.

- The hyperparameter tuning is rather intense for theoretically inclined audiences. It would be of great service if the paper can report the experimental results of a simple (not the best) hyperparameter setup for easy repetition/reproduction, in conjunction with these more tuned results. The paper should also specify the data processing step.

Questions:

- Does the $\pi$-parameterization inherit most properties of the $\mu$-parameterization, other than being different in its limit’s analytical form and training procedure?

    For example, the $\mu$-parameterization has a somewhat paradoxical property (despite its name "maximal") that $\Delta w(t) / w(0) \to 0$ as width tends to infinity in most layers, for $\Delta w(t)$ the gradient update at time $t$ and $w(0)$ the initial weight. That is, each weight coordinate moves very little from its initialization. Does the $\pi$-parameterization share the same property?

- How would one initialize the weights in $\pi$-nets if the widths are not all equal? Here the same projection $\Omega$ is used, restricting the widths to be all equal.

- Footnote 10 seems to say that doing gradient accumulation on $\pi$-nets could be difficult due to smoothness of $\mathcal{V}_\phi$. Is there such difficulty with the computation of the $\pi$-limit?


**Summary Of The Paper:**

The paper proposes an infinite-width parameterization ($\pi$-parameterization) that is similar to $\mu$-parameterization of Yang and Hu, with two important tweaks: the initialization involves a random projection of a certain form and the gradient update invokes the same projection. This results in an infinite-with description that is more amenable to computation and analysis. The paper demonstrates experimentally that this method achieves feature learning and better performance than NTK.

**Summary Of The Review:**

The proposal in the paper is interesting and novel, but the paper has deficiencies in its research methodology. It would be in a much better shape if several more fundamental issues are addressed.

---

> ### Author Response · Authors · 2021-11-14
> **(1) New plots to answer main concerns**
>
> We are grateful for the reviewer's thoughtful feedbacks.
>
> > The paper does not demonstrate convincingly if a finite-width net under -parameterization approaches the -limit as widths go to infinity; that is, a verification of Theorem 3.5.
>
> We have added plots verifying the convergence of the entire loss curve of a finite pi-net to that of the pi-limit. See Fig 9 and 10. Indeed, one of the first things we did in our research was developing programs to automatically make these kinds of plots as they are very useful for debugging our implementations, akin to gradient checking for autograd. We apologize for forgetting to include these plots in our initial submission. We have always intended to include such programs in our final code release.
>
> > Figure 5 looks concerning: as it shows, a width of 40,000 (!) still has a big gap to the limit on CIFAR10. I think this warrants deeper investigation.
>
> This is actually expected because cifar10 has 50k training examples, so the feature kernel of even width 40k (which has rank 40k) is still rank deficient. Indeed, you can see there's already a gap at initialization (where the limit is obviously correct). So you are just pointing out the slowness of convergence of kernel regression accuracy with width in a kind of random feature model.
> Even at initialization, the width needs to be quite large compared to the data size (= kernel size) before we see visible convergence of kernel regression (KR) accuracy. This is just because KR is sensitive to small eigenvalues of the kernel, which converges quite slowly with width compared to the largest eigenvalues (the optimal ridge in our experiments are all quite small that we do not expect them to significantly help convergence).
>
> To prove this point, we verify that if we just restrict ourselves to a smaller subset (400 training examples) of CIFAR10, then we see visible convergence of FKR accuracy at initialization. This is included as figure 11 in the new revision.
>
> > From the experiments, r should be large. But the need for large r complicates the studies.
>
> Actually, our experiments show that r does *not* need to be large before the pi-limit outperform mu-net (or standard parametrized finite networks). As clearly stated in the table 1 captions, the largest r for our main results is 400, but the width is 2048. A priori, one perhaps should bet that pi-net and pi-limit should severely underperform mu-net, since preactivations of pi-net and pi-limit live in a 400-dimensional space but those of mu-net live in a 2048-dimensional space. Nevertheless, pi-limit outperforms mu-net and (and pi-net is competitive with mu-net) in all settings.
>
> > Firstly this questions the relevance of the -limit itself, if one takes it as a theoretical limit of -net. In particular, -limit is derived with  while keeping  constant. Once  is a nontrivial fraction of the width, one should expect some nontrivial random matrix effect to kick in and the -nets to have nontrivial deviation from the -limit, casting doubt on the validity of -limit. This is indeed the case in the experiments.
>
> Again, all of our main experiments have small r compared to width. Fig 4(Middle,Right) are there to simply answer natural questions regarding the effect of r on the performance of pi-net and pi-limit that we thought a typical reader would wonder about; but the large r experiments did not contribute to our main table.
>
> Certainly, as you point out, if r is of the same size as width, then there is reason to doubt whether the limit is relevant. But our point is that a pi-limit or a pi-net with small r compared to width can match or outperform standard or mu-nets and demonstrate feature learning --- that is, our main claim is an *existence statement* (i.e. “there exists an interesting regime (the small r regime) worth studying”) rather than a *for-all statement* (e.g., “for any interesting regime (such as the large r regime), the pi-limit accurately captures it”). We believe our discovery of this small-r regime is unexpected by and is of interest to the deep learning theory community.
>
> Finally, right above section 5, we actually motivate the future study of a joint limit where r and width both go to infinity at a fixed ratio --- but likely this limit is not amenable to efficient calculation on real datasets like the pi-limit, so is outside of our main focus here.
>
> > Secondly if the proposal is to consider -limit as a new neural architecture with a specific training algorithm ...
>
> We do not propose this in this work. It is strictly intended as a new theoretical limit that we believe captures feature learning behavior in practical networks and can replicate their performance, and thus can be beneficial in advancing theoretical understanding of neural networks.

---

> > ### Author Response · Authors · 2021-11-17
> > **Have we addressed the main concerns?**
> >
> > Hi reviewer emnb,
> >
> > Please let us know if our response has been satisfactory. We are happy to work with you to address any follow-up questions if there are any. Otherwise, please consider raising your score.
> >
> > Thanks!

---

> > > ### Comment · Reviewer_emnb · 2021-11-17
> > > **reply**
> > >
> > > Many thanks to the authors for the extensive reply!
> > >
> > > I have a quick glance and will update my review in the coming days, but there is one request I should ask now. In figures 9 and 10, $r=2$ is used. This is very small compared to the typical value of $r$ used throughout the paper. Do you have the data for larger $r$ (like $r=400$)? I think it is more informative to plot something with the ratio $r/width$ closer to the typical value in the paper. Besides if there is the data for the training accuracy or test accuracy, it would be better to plot it on the same figure since it's easier to interpret these metrics.
> > >
> > > If the data to plot is not available and it is not possible to obtain it in time, I will try to make a judgement call based on what's available. A rule of thumb in statistics is that one should be very wary of data matrices of an aspect ratio > 1/5; they are more likely to have high-dimensional behaviors and deviate from classical statistics. As such, without clear evidence, it's questionable if $r=400$ is anything small compared to a width of 2048.

---

> > > > ### Author Response · Authors · 2021-11-18
> > > > **New figure 12 added for r=400**
> > > >
> > > > Our pleasure!
> > > >
> > > > We have uploaded a new revision with the plot (figure 12) you requested of r=400 for both training loss and accuracy. Let us know what you think :)

---

> > ### Comment · Reviewer_emnb · 2021-11-22
> > **reply**
> >
> > Thanks again for the replies so far!
> >
> > I will update my review and scoring (if there is a change) after discussing with other reviewers. Here are a few notes on specific replies from your end:
> >
> > - Thanks for updating with figure 12. It is now clearer to what extent the $\pi$-limit describes various $\pi$-nets at different widths and at a reasonably large $r=400$. On one hand, one can observe nontrivial deviations from the $\pi$-limit for the $\pi$-net of width 2048, and at times, that of width 32768 (which I think is very large in the context of CIFAR10 and certainly not among the typical widths that were used throughout the paper). Though some of these deviations could be due to discontinuity of the accuracy metric, I think the overall picture shows that the $\pi$-limit does not serve as a full substitute for a $\pi$-net at a reasonable width; this deviation due to finite width effects will be interesting for further studies. On the other hand, it is nice to see that the qualitative feature of the $\pi$-limit is followed by the $\pi$-nets.
> >
> > - I understand that your footnote 12 is merely a statement that practical networks tend to have exhaustive hyperparameter search done. I agree that the maximization over hyperparameters and the infinite-width limit do not necessarily commute. What I meant to say is that $\mu$-nets and standard-parameterized nets are expected to have qualitative differences under sufficiently large widths, unless the hyperparameter search is done very carefully to cover the gap. Since we already zoom into the details, one thing to notice is that $\mu$-nets have non-uniform learning rates applied to different layers (if one is to view them as "standard-parameterized" nets), and specifically the learning rates are scaled either up or down with the widths differently for different layers. As far as I understand, this non-uniformity is important for the $\mu$-limit. With width $2^{11}$, I'm afraid that the hyperparameter search as done in the paper does not compensate for this non-uniformity in different scalings with the widths of the learning rates. This point is not the focus of the study anyway.
> >
> > - A request: can the authors later (after this period) specify how the standard deviations of the test accuracies are estimated and add them to the tables? Actually the suggestions from Reviewer pNHj to find setups where the difference could be amplified in a more convincing way are good ones, if the authors have sufficient time after this period.

---

> > > ### Author Response · Authors · 2021-11-23
> > > **Thanks for the discussion!**
> > >
> > > We really appreciate your suggestions and how engaged you are :) This discussion has been fun (hopefully it was to you as well)!
> > >
> > > -------------------------------------------
> > >
> > > > Though some of these deviations could be due to discontinuity of the accuracy metric, I think the overall picture shows that the -limit does not serve as a full substitute for a -net at a reasonable width; this deviation due to finite width effects will be interesting for further studies. On the other hand, it is nice to see that the qualitative feature of the -limit is followed by the -nets.
> > >
> > > We agree. But just to be clear, on the same dataset and with the same architecture, a network in NTK parametrization with width 2048 would exhibit similar level of deviation from the infinite-width NTK limit (we are happy to add a plot of this in the final version). So the pi-limit is not an unreasonable limit in comparison to established works.
> > >
> > > > With width 2^11, I'm afraid that the hyperparameter search as done in the paper does not compensate for this non-uniformity in different scalings with the widths of the learning rates.
> > >
> > > From the beginning, we were aware of the point you bring up, so we actually first did a coarse hyperparameter search (that would cover this difference in non-uniformity) to determine the finer grids we listed in the paper. We observed that the optimal hyperparameter ranges were stable in the mu-parametrization (but not the standard parametrization), which justified why we went with mu-nets instead of standard nets as the baseline. Note this is consistent with [recent findings](https://openreview.net/pdf?id=Bx6qKuBM2AD) that showed optimal hyperparameters are stable with width under $\mu$P but not SP (e.g., fig 3 in that paper specifically shows this for MLPs) and verified this on models up to GPT-3. In any case, we are happy to give more details on our first round coarse search if this alleviates your concerns.
> > >
> > > > A request: can the authors later (after this period) specify how the standard deviations of the test accuracies are estimated and add them to the tables?
> > >
> > > Of course.
> > >
> > > > Actually the suggestions from Reviewer pNHj to find setups where the difference could be amplified in a more convincing way are good ones, if the authors have sufficient time after this period.
> > >
> > >
> > > While we agree with the general sentiment here, we have concerns about the specific paper suggested by Reviewer pNHj as we stated in response.
> > > - If the intention is reaching SOTA, then this paper does not help because it is nowhere near 90+%.
> > > - If the intention here is to show that on the *vanilla* CIFAR10 setting there is a large gap, then it does not help either because it involves an uncommon pretraining + finetuning + conjugate gradient training procedure (see their section 4.4).
> > > - If being "vanilla" is not the point, then our Imagenet pretraining experiment already shows a large performance improvement over NTK.
> > > - If "CIFAR10" is not the point, then our Omniglot (which, in our opinion, is a *much much* better dataset to assess feature learning, and we hope the community will eventually agree) experiment already shows an enormous performance improvement over NTK.
> > >
> > > Now, we noticed that most of the improvement in Reviewer pNHj's suggested paper comes from data augmentation (flips, rotation, shift). We are happy to explore this as an additional "nonvanilla" setting peer to our Imagenet pretraining experiment (without other gadgets from that paper to keep our experiment as clean as possible).

---

> ### Author Response · Authors · 2021-11-14
> **(2) Efficiency, CIFAR gap, SP vs muP, LR drop**
>
> > The claim (in the paper title) is that -limit can be efficiently computed, but again there is little discussion. In particular, can the paper provide a snapshot of the runtime for a reasonable, easily reproducible setup?
>
> As we clearly state throughout the paper (e.g., in the abstract), by “efficient” we explicitly mean compared to the mu-limit, which can be exponentially hard to calculate. We have never claimed that the pi-limit is faster than SGD on a finite neural network. Just as NTK kernel regression is slower than training a finite network, so is the pi-limit.
>
> Nevertheless, in the new revision, table 8 and 9 describe the runtime of our networks on CIFAR10 and Omniglot. The pi-limit can be just slightly slower than the mu-net (Omniglot, 1-hidden-layer) to more than 20 times slower (CIFAR10, 4-hidden-layer).
>
> In contrast, the mu-limit (for relu) would not even have an easy-to-evaluate exact formula for beyond 1 step of SGD; even if one evaluates it via Monte-Carlo, one would not be able to compute beyond a few steps of SGD if we require accuracy comparable to the pi-limit formulas.
>
> If the reviewer does not like the word "efficient", we are happy to hear suggestion for a replacement.
>
> > The paper claims outperformance of -limit and -nets over others in Table 1, but some of these numbers look too close to make a call. An increase of 2% for CIFAR10 could be borderline (given the nontrivial randomness in the generation of the matrices  and ).
>
> The CIFAR10 gap is not small. In fact, prior works such as [Lee et al. 2018](https://openreview.net/forum?id=B1EA-M-0Z) and [Lee et al. 2020](http://arxiv.org/abs/2007.15801), widely cited for saying NNGP/NTK outperform corresponding MLPs, based their claims on the same margins (2%). It would be strange if we are not allowed to correct their claims by the same margin. In addition, the standard deviations of the pi-limit, pi-net, and mu-net accuracies on CIFAR10 due to initialization of $A^l$ and $B^l$ are all around 0.2%, so the 2% margin represents a 10-sigma event and thus is very statistically significant.
>
> > Footnote 12 is quite concerning: it is well-known that standard-parameterized MLPs should behave differently from -limit (and are close to NTK at best) as widths tend to infinity, but this footnote seems to suggest that the experimented -nets may operate in a regime far from the -limit?!
>
> Do you mean mu-net and mu-limit? We are just stating the trivial mathematical fact that muP and SP differ only by the hyperparameters of learning rate, weight multipliers, and initialization variance. When we let width vary, they differ by powers of width. But, *for any fixed width*, such differences "become constants" and can be tuned away.
> In other words:
>
> $$\max_{hyperparams} accuracy(\text{width-$n$ $\mu$-net}) = \max_{hyperparams} accuracy(\text{width-$n$ standard net}),  \text{ for any \emph{fixed} $n$}$$
> where $hyperparams$ consists of learning rate, weight multipliers, and initialization variance.
> (This equality holds true for any pair of abc-parametrizations).
> This max accuracy, shared between mu-net and standard net, is what we report in table 1, where width is fixed to 2048.
>
> On the other hand, if we *fix these hyperparameters* and increase width, then muP and SP would arrive at different limits, which is what we believe you are trying to say. This is not what we are doing with the finite neural networks.
>
> In some sense, you may be confused about the order of $\lim_{n\to\infty}$ and $\max=\max_{hyperparams}$, which don't necessarily commute. Our mu-net results can be thought of as approximating
> $$
>    \lim_{n\to\infty} \max accuracy(\text{width-$n$ $\mu$-net})$$
> $$
> = \lim_{n\to\infty} \max accuracy(\text{width-$n$ standard net})$$
> by setting $n=2048$, as discussed above, but we cannot commute $\lim$ and $\max$ at will
> $$
> \ne \max  \lim_{n\to\infty} accuracy(\text{width-$n$ standard net})$$
> which, as you said,
> $$
> \approx \max accuracy(\text{NTK}).$$
>
>
> Please let us know if this clears up your confusion.
>
> > Footnote 13 seems unfair. Firstly the design of the learning rate schedule should be considered as a knob in hyperparameter tuning. Secondly the outcome of convex optimization would differ if one switches from a constant learning rate schedule to one with learning rate drops.
>
> We strongly disagree, especially with your second point. Doing kernel gradient descent (with a positive definite kernel) for an infinite amount of time (which is equivalent to kernel regression) would attain exactly the same results whether one has a single learning rate drop at epoch 30 or not, just because the optimum is unique. On the other hand, we only train finite networks (and pi-limit) for a finite amount of time, so without learning rate drop they are in fact disadvantaged unfairly, as well as being far from practice.

---

> ### Author Response · Authors · 2021-11-14
> **(3) Reproducibility, mu-limit vs pi-limit, varying widths, grad acc vs grad cat**
>
>
> > It would be of great service if the paper can report the experimental results of a simple (not the best) hyperparameter setup for easy repetition/reproduction, in conjunction with these more tuned results. The paper should also specify the data processing step.
>
> We will in fact go beyond the reviewer's suggestion and on publication release our code that allows anyone to train a pi-limit as easily as a finite neural network, along with hyperparameters and data processing steps to reproduce our results.
>
> > For example, the -parameterization has a somewhat paradoxical property (despite its name "maximal") that  as width tends to infinity in most layers, for  the gradient update at time  and  the initial weight. That is, each weight coordinate moves very little from its initialization. Does the -parameterization share the same property?
>
> In the mu limit, this property is because the random iid weights $W_0$ at initialization behaves differently than the “outer product” matrix $\Delta W$ coming from the gradients, so they must be scaled differently to ensure they contribute equally (in terms of scaling with width) in the matmul $(W_0 + \Delta W)x$. In the pi limit, because of the pi-initialization, the initial matrices $W_0$ also are “outer product” matrices by construction, so the coordinate size of the gradients is of the same order as that of the initial weights.
>
> > How would one initialize the weights in -nets if the widths are not all equal? Here the same projection  is used, restricting the widths to be all equal.
>
> This is easily done by using differently shaped, independently sampled Omega for different *hidden layers*, by which I mean the "junction" between consecutive weight matrices.
> More precisely, you would sample $\Omega^l \in \mathbb{R}^{n_l \times r}$ for each hidden layer $l=1, \ldots, L$ with width $n_l$.
> Then you generate weights of the middle layers by
> $$
> w^l \gets \frac 1 n \Omega^l A^{l\top} \phi(B^l \Omega^{l-1\top}) \in \mathbb{R}^{n_l \times n_{l-1}}
> $$
> (notice how $w^l$ depends on $\Omega^l$ and $\Omega^{l-1}$),
> and for input and output layers
> $$
> w^1 \gets \frac 1 {\sqrt n} \Omega^1 A^{1 \top} \in \mathbb R^{n_1 \times d}; \quad
> 	w^{L+1} \gets \frac 1 {\sqrt n} A^{L+1\top} \phi(B^{L+1}\Omega^{L\top}) \in \mathbb R^{d_{out} \times n_{L}}
> $$
>
>
> > Footnote 10 seems to say that doing gradient accumulation on -nets could be difficult due to smoothness of . Is there such difficulty with the computation of the -limit?
>
> Just to be clear, footnote 10 says that if one were to naively train a *pi-limit* (not a pi-net) by gradient accumulation on A and B rather than concatenation (as the math would tell us), then one runs into difficulties because of the smoothness of V_phi.
> One can intuit that this is because directly optimizing A and B using gradient accumulation means that one is doing a local search in a neighborhood of the (A, B) space that induce nearly linear f.
> However, training the pi-limit via grad concatenation does not encounter such issue. One can intuit that this is because this procedure corresponds to (projected) SGD in a finite network, where we are doing local search in a neighborhood of the parameter space that contain highly varied, nonlinear functions.

---

### Decision · Program_Chairs · 2022-01-20

**Decision:**

Accept (Poster)

**Comment:**

This paper studies deep non-linear infinite-width neural networks that go beyond the NTK and learn features. This paper extends the prior result on shallow neural networks to deep neural networks and empirically evaluates the deep inf-wide nn. The reviewers find the contributions in the paper valuable. The meta reviewer agrees and thus recommends acceptance.